

# Probabilistic deconstruction of a theory of gravity, part II: Curved space

Sunok Josephine Suh⋆

Kavli Institute for Theoretical Physics, University of California,
Santa Barbara, CA 93106-4030, U.S.A.

⋆ sjsuh@kitp.ucsb.edu

## Abstract

We propose that the underlying context of holographic duality and the Ryu-Takayanagi formula is that the volume measure of spacetime is a probability measure constrained by quantum dynamics. We define quantum stochastic processes using joint quantum distributions which are realized in a quantum system as expectation values of products of projectors. In anti-de Sitter JT gravity, we show that Einstein's equations arise from the evolution of probability under the quantum stochastic process induced by the boundary, with the area of compactified space in the gravitational theory identified as a probability density evolving under the quantum process. Extrapolating these and related results in flat JT gravity found in [SciPost Phys. 15, 174 (2023)], we conjecture that general relativity arises in the semi-classical limit of the evolution of probability with respect to quantum stochastic processes.

# 1 Introduction

Over the past few decades, there have been some significant clues as to the question of how quantum theory and gravity can be brought together. Holographic duality [1–3]—positing a dual relationship between certain quantum systems and gravitational theories in anti-de Sitter space with one more spatial dimension—cast the problem of quantum gravity, at least in a certain class of examples, into one of decoding the relationship that exists between a non-gravitational quantum system and its gravity dual. The Ryu-Takayanagi [4, 5] formula shed light on an important aspect of this relationship, which is that the gravitational description of the quantum system is intimately connected to its information content.

In light of the fact that the Ryu-Takayanagi formula is a relation that holds between a special (extremal) value of the codimension-two area in the gravitational theory, and an informational quantity (quantum entropy) calculated from the density matrix of the quantum system, a natural question that has been lurking all along is what problem the gravitational equations of motion—or Einstein's equations—themselves are solving relative to the quantum system, such that a relation like the Ryu-Takayanagi formula could hold. Given the larger context described above, to answer this question amounts to directly tracing the quantum origin of spacetime and gravity in the setting of holographic duality; one could hope to extract from a consistent answer a framework general enough to extend beyond gravity in anti-de Sitter space. Put another way, we would like to "deconstruct" gravity, i.e. understand it in a sufficiently new way such that it is subsumed as part of quantum theory, expressing some particular aspect of quantum theory that we have not been able to articulate before.

In this paper, which is a continuation of the preceding [6], we propose a general solution to this problem, presenting and extrapolating a complete analysis in the simple example of Jackiw-Teitelboim (JT) gravity [7–9], in cases of both zero and negative cosmological constant. We propose that the volume measure of spacetime in general relativity is in fact a *probability measure*[1] constrained with respect to a *quantum stochastic process*, with the latter given by a sequence of *joint quantum distributions* over time governing a quantum observable. We propose that Einstein's equations arise in the semi-classical limit of an exact *generator equation* for the quantum stochastic process, which solves for probability measures consistent with evolution with respect to joint quantum distributions of the process.

Our proposal for defining a quantum stochastic process is grounded in an analysis of certain dynamical correlators that can be computed for an observable of a quantum system: *expectation values of products of projectors* (EVPP's), of the form

$$q_{T_1}(x_1) = \text{Tr}\left(\rho\, e^{iHT_1} P(x_1) e^{-iHT_1}\right), \quad q_{T_2,T_1}(x_2, x_1) = \text{Tr}\left(\rho\, e^{iHT_1} P(x_1) e^{-iH(T_1-T_2)} P(x_2) e^{-iHT_2}\right), \quad \dots \tag{1}$$

---

[1] We will use the term probability measure to refer to measures defined in the context of probability theory (as opoosed to e.g. volume measures in general relativity) even when they do not integrate to 1, and reserve the term probability *distribution* for probability measures that do integrate to 1.

where $x_i$ are eigenvalues of the operator $X$ corresponding to the observable, and $P(x_i)$ is the projection operator onto the eigenspace of the Hilbert space with eigenvalue $x_i$.[2,3] Let us first clarify that although the EVPP's involve projection operators, no measurement or collapse of the quantum state is involved; the projectors are inserted only on one side of the density matrix. The correlators can be viewed as quantum generalizations of joint probability distributions defining a classical stochastic process, in the following sense. The $q^{(n)} \equiv q_{T_n,\ldots,T_1}(x_n,\ldots,x_1)$ each sum to one and satisfy marginalization relations, properties following from the completeness of projectors and $\text{Tr}(\rho) = 1$. However, for $n \geq 2$, they are generically complex rather than positive, due to non-commutativity between the projectors at different times. We propose to abstract these attributes from the EVPP's of actual quantum systems and view them as defining a set of *joint quantum distributions* for an observable, which in turn define a quantum stochastic process. Then such a process can be defined without needing to reference some quantum system and its associated Hamiltonian and density matrix, a feature that allows it to be relevant to quantum gravity beyond the anti-de Sitter setting.

Now, an important question which subsequently arises, is in which cases and how contact can be made with joint probability distributions which are positive, starting from a quantum stochastic process. This question is similar in nature to asking how contact can be made with chaos starting from a quantum system [10–13]. Recall chaos is a dynamical phenomenon that could be intrinsically defined only for classical systems; exponential divergence of trajectories in phase space is a concept that only makes sense in the context of a continuum phase space that is infinite. In trying to make contact with this notion starting from quantum systems, it was found to be useful to examine a certain class of dynamical quantum correlators, out-of-time-order correlators (OTOC's), and to take the semi-classical limit [10, 11].

In our case, we also have an intrinsically classical dynamical phenomenon we would like to make contact with in quantum systems: a stochastic process with positive joint probability distributions. The joint quantum distributions we have defined in (1) and below are analogous to OTOC's in that they are dynamical quantum correlators which will be useful in this regard. In similar spirit to the case of chaos, we propose that positive joint probability distributions may be extracted from the semi-classical limit of joint quantum distributions, as follows. If possible values of the observable of a quantum stochastic process are sufficiently dense in its target space so that we may consider continuum joint quantum distributions $q_{T_n,\ldots,T_1}(dx_n,\ldots,dx_1)$ which are invariant over infinitesimal volumes, and if there exists a semi-classical limit of the joint quantum distributions, we can identify *effective* joint probability distributions $p_{T_n,\ldots,T_1}(dx_n,\ldots,dx_1)$ in the strict classical limit, via leading saddle-point evaluations of total integrals of joint quantum distributions over target space:

$$1 = \int_{x_1,\ldots,x_n} q_{T_n,\ldots T_1}(dx_n,\ldots,dx_1) \underset{\text{leading sadd. pt. eval.}}{\approx} \int_{x_1,\ldots,x_n} p_{T_n,\ldots T_1}(dx_n,\ldots,dx_1). \qquad (2)$$

This is because in the leading saddle-point approximation, an integral is evaluated along a path of constant phase, so one can identify an effective positive integrand for a unit integral.

What is the connection to gravity of this contact we can make with a classical dynamical phenomenon? In the case of chaos, once we know to how to make contact with it in quantum systems, gravity can be shown to be associated with an extremal limit of maximal chaos, as quantified by a maximal Lyapunov exponent [14]. Similarly, in the case of stochastic processes, it seems there is an extremal limit of a stochastic process being locally Markov, that could be associated to gravity emerging from quantum systems. Let us try to explain the connection between Markovianity and gravity we have seen.

---

[2]For brevity we have written expressions assuming the Hamiltonian is time-independent.

[3]We are interested in observables broadly defined to include more than Hermitian operators, e.g. a normal operator with eigenvalues in the complex plane rather than on the real line.

In the case that a classical stochastic process has the Markov property, there is a sense in which the action of any conditional probability $\mu_{T_2,T_1}(dx_2; x_1) = p_{T_2,T_1}(dx_2, dx_1)/p_{T_1}(dx_1)$ over finite time on a measure $\nu(dx_1)$ can be obtained as the exponential of an infinitesimal generator. The latter expresses the instantaneous time derivative of the action, and completely characterizes the process, including higher-order joint probability distributions. (In the following, our explicit discussions will be related to the linear problem where conditional probabilities including the kernel $\mu_{T_2,T_1}$ do not depend on the measure they are acting on. We will also mostly consider homogeneous processes for which joint probabilities and the kernel $\mu_{T_2,T_1} = \mu_{T_2-T_1}$ only depend on time differences.) Furthermore, if the process is sufficiently "macroscopic", meaning the probability kernel $\mu_T(dx_2; x_1)$ localizes in target space at small times, the generator can be related to spatial derivatives of finite order in target space. This relation is expressed by the well-known Fokker-Planck equation solving for probability measures consistent with the stochastic process. See Section 3 of [6] for elaboration.

As we will describe in Section 2, it is in fact possible to associate generators to a non-Markovian process, such that they generate a local, Markovian approximation to the actual process.[4] Then given (2), we may postulate that a quantum stochastic process involving a large number of degrees of freedom and having a semi-classical limit, and which is *locally Markovian* to a sufficient degree,[5] is characterized by a *quantum generator equation* expressing the instantaneous time derivative of the action of its conditional *quantum* distribution

$$\kappa_{T_2,T_1}(dx_2; x_1) = \frac{q_{T_2,T_1}(dx_2, dx_1)}{q_{T_1}(dx_1)}, \tag{3}$$

on probability measures.

Let us write down such a generator equation assuming we have a linear problem where the measure $\nu(dx)$ we are solving for does not itself enter the joint quantum distributions $q_{T_n,\dots,T_1}(dx_n,\dots,dx_1)$, and instead a sub-measure $\mathcal{D}x$ which is non-dynamical factors out of both as $\nu(dx) = \mathcal{D}x\,\Phi(x)$ and $q_{T_n,\dots,T_1}(dx_n,\dots,dx_1) = \mathcal{D}x_1\cdots\mathcal{D}x_n\,q_{T_n,\dots,T_1}(x_n,\dots,x_1)$:

$$\lim_{T_{21}\to 0^+}\int \mathcal{D}x_1\,\frac{\partial_{T_2}q_{T_2,T_1}(x_3,x_1)}{q_{T_1}(x_1)}\Phi(x_1) = \lim_{T_{21}\to 0^+}\lim_{T_{32}\to 0^+}\frac{1}{T_{32}}$$
$$\times\left(\int \mathcal{D}x_1\mathcal{D}x_2\,\frac{q_{T_3,T_2,T_1}(x_3,x_2,x_1)}{q_{T_1}(x_1)}\sum_{|\boldsymbol{k}|=0}^{\infty}\frac{\Phi^{(k)}(x_{13}=x_{12})}{\boldsymbol{k}!}(\boldsymbol{x_2}-\boldsymbol{x_3})^{\boldsymbol{k}} - \int \mathcal{D}x_1\,\frac{q_{T_2,T_1}(x_3,x_1)}{q_{T_1}(x_1)}\Phi(x_1)\right). \tag{4}$$

Note we have denoted $T_{ji} = T_j - T_i$ and in taking $T_{21} \to 0$, we are taking the instantaneous limit of the action of $\partial_{T_2}\kappa_{T_2,T_1}$ on some arbitrary probability density $\Phi(x)$.

The above is precisely the situation that applies to the quantum stochastic process describing the "position" observable of the boundary of AdS JT gravity. The target space of the process is two-dimensional anti-de Sitter space with a non-fluctuating measure $\mathcal{D}x = dx\sqrt{-g}$, over which we can consider some probability density $\Phi(x)$. We obtain the joint quantum distributions of the process using EVPP's in the quantum theory of the boundary [15], in a thermal state and with parameters in an appropriate limit. The quantum stochastic process corresponding to flat JT gravity can be obtained by taking their asymptotic limit at short distances.[6] What we find in these cases, is that Einstein's equations of JT gravity are reproduced as components of the above generator equation at leading non-vanishing order in the semi-classical limit, with the probability density $\Phi(x)$ identified as the dilaton field or area of compactified space in JT

---

[4]Generically the Markovian approximation only replicates single-event distributions of the original process.

[5]Making the condition of local Markovianity precise is a problem we leave for the future.

[6]We compare all time/length scales to the AdS radius, set to 1.

gravity! In other words, (4) reduces to

$$\underset{\text{semi-classical limit}}{\Longrightarrow} \qquad \lim_{x_1 \to x_3} \frac{\partial X^\mu}{\partial l} \frac{\partial X^\nu}{\partial l} \left( \nabla_\mu \nabla_\nu - g_{\mu\nu} \nabla^2 - g_{\mu\nu} \Lambda \right) \Phi(x_3) = 0 \,, \tag{5}$$

where $l(x_3; x_1)$ is an appropriately scaled geodesic distance to $x_3$ from another point $x_1$ at time-like separation, and $\Lambda$ is the cosmological constant. (In $\text{AdS}_2$ we have set the radius of curvature to be one so that $\Lambda = -1$.)

Normally, we would have derived the Einstein's equations appearing as components of (5) by varying the bulk action of JT gravity $I_{\text{JT}}[g, \Phi] = \frac{1}{4\pi} \int_{\mathcal{M}} d^2x \sqrt{-g} \, \Phi(R - 2\Lambda)$ with respect to the metric.[7] Here, we have reconstructed the equations by considering an informational and dynamical phenomenon in quantum theory, that of a quantum stochastic process constraining probability, and identifying the constrained probability measure with the volume measure of spacetime!

In [6], we showed that the asymptotic quantum stochastic process of AdS JT gravity at short time scales, corresponding to flat JT gravity, is exactly Markovian in the classical limit—i.e. the joint probability distributions it produces via (2) have the Markov property. Furthermore, we showed that the flat asymptotics of joint quantum distributions in the generator equation (4) lead to the derivative terms in (5). The full quantum stochastic process of AdS JT gravity is no longer Markovian in the aforementioned sense. However, it is still apparently sufficiently Markovian in a local sense so that it can be characterized by a generator equation. As we will show in this paper, we can recover the full Einstein's equations of AdS JT gravity including the cosmological constant, by employing in (4) joint quantum distributions resulting from Schwarzian dynamics at long time scales [15, 16].[8] Roughly, this is possible because proper time is renormalized in the regularized quantum theory of the boundary in which we compute joint quantum distributions, and we can go to small renormalized (proper) times while staying at long time scales, then interpolate to vanishing renormalized times as is necessary for the generator equation.

It is natural to extrapolate the above results, and conjecture the following: general relativity arises in the semi-classical limit of the evolution of probability with respect to quantum stochastic processes, with the volume measure of spacetime being a probability measure in the target space of a quantum observable, evolving with respect to the stochastic process governing the observable. We anticipate that for a quantum stochastic process to give rise to gravity in this way, besides having possible values of the observable that are sufficiently dense in a multi-dimensional target space, and joint quantum distributions with a semi-classical limit, it should be Markovian in a local sense so as to be characterized by a generator equation.[9] Generically, in contrast to the linear case of JT gravity, the probability measure $\nu(dx) = \sqrt{-g} \, dx$ one is solving for will simultaneously enter the joint quantum distributions of the process and evolve under them, so that the generator equation is non-linear. We extrapolate that Einstein's equations, also non-linear in the general case, will arise as components of such a generator equation in the leading non-vanishing order in the semi-classical limit.

Sections in the rest of the paper are organized as follows: in Section 2, we give a proof of concept that one can associate generators to a non-Markov process which give a local, Markovian approximation to the actual process. In Section 3, we study the geometry and dynamics relevant to the quantum stochastic process induced by the boundary of JT gravity at long time scales, ultimately reconstructing Einstein's equations from the generator equation of the process, with the area of compactified space in the gravity theory identified as a probability

---

[7]The equations solve for $\Phi$, a scalar field corresponding to the area of compactified space at a point. Meanwhile, the two-dimensional metric $g$ is fixed by varying with respect to $\Phi$.

[8]The cosmological constant term in (5) is produced by the three-event joint quantum distribution in (4).

[9]We speculate that Markovianity is related to flatness of the associated spacetime, and the condition of local (sufficient) Markovianity, with the locally flat characterization of spacetime manifolds.

density evolving with respect to the process. In Section 4, we discuss various aspects of our results as well as future directions for research.

## 2 Generators of a non-Markovian process

Here we work in the context of classical stochastic processes, and extract an observation from [17]: that it is possible to associate generators to a non-Markovian process, such that they generate a local, Markovian approximation to the actual process. (For a review of basic concepts in probability theory that provide appropriate background, see Section 3 of [6].)

Recall that a Markov process is characterized by the semi-group property of its conditional probabilities $\mu_{T_2,T_1}(dx_2; x_1) = \mathbb{P}(X_{T_2} \in dx_2 | X_{T_1} = x_1)$,

$$\mu_{T_3,T_1}(dx_3; x_1) = \int \mu_{T_3,T_2}(dx_3; x_2) \mu_{T_2,T_1}(dx_2; x_1). \tag{6}$$

The semi-group property implies that a generator can be defined which exponentiates to the operator on measures induced by a probability kernel, $(M_{T_2,T_1} \nu)(dx_2) = \int \mu_{T_2,T_1}(dx_2; x_1) \nu(dx_1)$; heuristically, $M_{T_2,T_1} \sim \mathcal{T} e^{\int_{T_1}^{T_2} dT\, G_T}$, $G_T = \lim_{T' \to T^+} \partial_{T'} M_{T',T}$.

For a non-Markovian process, the semi-group property fails,

$$\begin{aligned} \mu_{T_3,T_1}(dx_3; x_1) &= \int \mu_{T_3,T_2,T_1}(dx_3; x_2, x_1) \mu_{T_2,T_1}(dx_2; x_1) \\ &\neq \int \mu_{T_3,T_2}(dx_3; x_2) \mu_{T_2,T_1}(dx_2; x_1). \end{aligned} \tag{7}$$

However, the authors of [17] made the observation that the equality (6) continues to hold as long as one is acting on a probability distribution on both sides, that is

$$\int \mu_{T_3,T_1}(dx_3; x_1) p_{T_1}(dx_1) = \int \int \mu_{T_3,T_2}(dx_3; x_2) \mu_{T_2,T_1}(dx_2; x_1) p_{T_1}(dx_1). \tag{8}$$

This implies that if we consider a Markov process generated by the instantaneous time derivatives of the conditional probabilities $\mu_{T_2,T_1}$ of the non-Markovian process, and specify its probability distribution at some time $p_{T_1}(dx_1)$ to be the same, its single-event distributions $p_T(dx)$ will always agree with those of the non-Markov process. We may view the Markov process thus constructed as a local approximation to the non-Markov process, in that we cannot distinguish the two processes as long as we are considering single events in time. The multi-event distributions $p_{T_n,\dots,T_1}(dx_n,\dots,dx_1)$, on the other hand, will generically not be replicated by the Markov approximation.

The above provides theoretical underpinning for the following possibility: If a non-Markovian process is locally Markovian in a sense such that the local Markov approximation we have described is more powerful than expected, the generator equation involving generators of the Markov approximation may be sufficient to characterize the entire process.

## 3 Quantum stochastic process in AdS JT gravity

We now consider the quantum stochastic process induced by the boundary of AdS JT gravity that we introduced in [6]. That is, the observable of the process is the position of the boundary

and takes values in $\widetilde{\mathrm{AdS}}_2$, and its joint quantum distributions are given by EVPP's evaluated in the quantum theory of the boundary,[10]

$$
\begin{aligned}
q_{T_n,\dots,T_1}(x_n,\dots,x_1) &= \mathrm{tr}\big(\rho\, e^{iHT_1}|x_1\rangle\langle x_1|e^{-iHT_1}\cdots e^{iHT_n}|x_n\rangle\langle x_n|e^{-iHT_n}\big) \\
&= \frac{\langle x_n|e^{-iHT_{n,1}}\rho|x_1\rangle\langle x_1|e^{iHT_{21}}|x_2\rangle\dots\langle x_{n-1}|e^{iHT_{n,n-1}}|x_n\rangle}{\mathrm{vol}(\widetilde{\mathrm{SL}}(2,\mathbb{R}))}\,.
\end{aligned}
\tag{9}
$$

The quantum theory is specified by the action of a particle with spin $\nu = -i\gamma$,

$$
S = \int dT \left(\frac{1}{2}g_{\mu\nu}\dot{X}^\mu\dot{X}^\nu + \gamma\omega_\mu\dot{X}^\mu\right).
\tag{10}
$$

See [6,15,18] for details.

Assuming a thermal quantum state with inverse temperature $L$, one finds that in the holographic limit where the two parameters $\gamma$ and $L$ are large, there is a precise renormalization scheme in which the particle sees flat space at short distances i.e. it tends to follow smooth, straight trajectories. (Note we set the AdS radius to be 1, and all comparisons of length and time scales are with the AdS radius.) In this holographic renormalization scheme, the renormalized inverse temperature is given by

$$
\beta = \frac{L}{\gamma}\,,
\tag{11}
$$

and the energy of a particle takes possible values

$$
E = \frac{s^2}{2}\,,\qquad s\geq 0\,,
\tag{12}
$$

with the density of states being given by

$$
\rho(E) = \frac{\sinh(2\pi s)}{2\pi^2}\,.
\tag{13}
$$

(The thermal density matrix is given by $\rho = \int dE\, Z_\beta^{-1}e^{-\beta E}\mathrm{P}_E$, with the density operator $\mathrm{P}_E$, $\mathrm{tr}(\mathrm{P}_E) = \rho(E)$ encoding the density of states.) Using the energy parameter (12), the holographic limit can be expressed as

$$
\gamma \gg 1\,,\qquad \gamma^2 \gg s^2\,.
\tag{14}
$$

In other words, small energies as parametrized above dominate the thermal ensemble for large inverse temperatures $L \gg 1$.

The holographic limit was called the Schwarzian limit in [15] but in retrospect was mislabeled, as a Schwarzian action describes the particle in this limit, but only at long time scales. The actual dynamical regime described by the Schwarzian, consisting of the quantum system in the holographic limit *and* at long time scales, will be important to us for the following reason.

It turns out that the propagators or two-point functions of the boundary particle have a closed-form expansion in the holographic limit (14), in large $\gamma$, only in two dynamical regimes associated with distinct asymptotic geometries obtained from $\widetilde{\mathrm{AdS}}_2$. See Figure 1. One is the regime at short time scales (corresponding to $T_b \ll 1$, or bare proper times much smaller than AdS radius) in which the particle sees flat space. The other is at long time scales or $T_b \gg 1$, in

---

[10]We will be working in a dynamical regime in which only one boundary rather than both boundaries of $\widetilde{\mathrm{AdS}}_2$ are relevant, and accordingly, there is no factor of $\frac{1}{2}$ in the trace defined in the quantum theory.

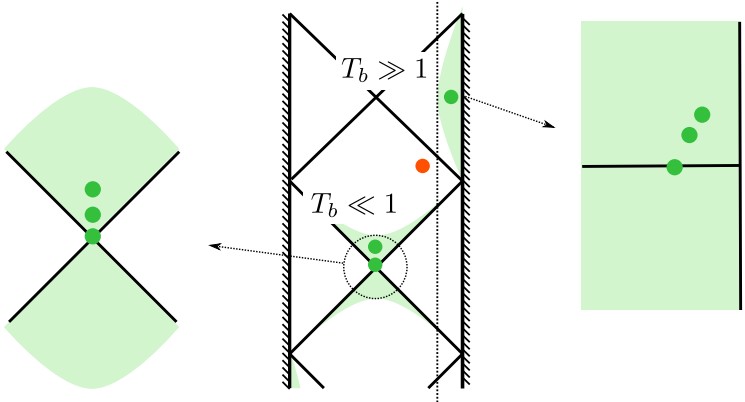

Figure 1: Taking the holographic limit (14), two-point functions of the boundary particle have a closed-form expansion in large $\gamma$ only at i) short time scales, when it sees flat space, ii) long time scales, when it sees the asymptotic near-boundary geometry of $\widetilde{\text{AdS}}_2$.

which the particle sees the asymptotic geometry near a boundary of $\widetilde{\text{AdS}}_2$. In [6], the quantum stochastic process induced by the dynamics at short time scales was analyzed from which we derived Einstein's equations of JT gravity in flat space. To see the cosmological constant, we must include corrections to the asymptotic process at short time scales, but we only regain analytic control of the dynamics at time scales much longer than the AdS radius, i.e. in the Schwarzian regime.

Fortunately, the fact that proper times in the boundary quantum system are renormalized as in (11) saves us. That is, the generator equation (4) involves taking renormalized (proper) times to zero, and we can go to short renormalized times $T = T_b/\gamma \ll 1$ while staying at long time scales $1 \ll T_b$, then interpolate to vanishing renormalized times $T \to 0$. This interpolation from long time scales corresponds to letting target points of the position observable approach each other in the asymptotic near-boundary geometry of $\widetilde{\text{AdS}}_2$, effective in the Schwarzian regime. See Figure 2. We will show that utilizing the geometry and dynamics of the quantum system in the Schwarzian regime as such, we can recover Einstein's equations of JT gravity in anti-de Sitter space with negative cosmological constant.

## 3.1 Geometry and dynamics in Schwarzian regime

Without loss of generality, we consider the asymptotic geometry $\mathcal{M}$ near the right boundary of $\widetilde{\text{AdS}}_2$, and a particle with spin $\nu = -i\gamma$, for which the classical trajectory goes up on $\mathcal{M}$. The metric on $\mathcal{M}$ can be obtained from taking the near-boundary form of the metric on $\widetilde{\text{AdS}}_2$—given by $ds^2_{\text{AdS}} = (-d\phi^2 + d\theta^2)/\cos^2\theta$ with $-\infty < \phi < \infty$ and $-\frac{\pi}{2} < \theta < \frac{\pi}{2}$—and using the scaled radial coordinate $\phi' = \gamma\left(\frac{\pi}{2} - \theta\right)$,[11]

$$ds^2_{\mathcal{M}} = \frac{-\gamma^2 d\phi^2 + d\phi'^2}{\phi'^2}\,. \tag{15}$$

Due to the scaling of the radial coordinate, the geometry in the radial direction is stretched out and light cones are flattened, see Figure 2.

In considering the quantum stochastic process of the boundary particle in the Schwarzian regime on $\mathcal{M}$, and ultimately computing the generator equation (4), it is necessary to study

---

[11]The notation $\phi'$ for the spatial coordinate is motivated by the fact that $\frac{d\phi}{dT} \approx \gamma\left(\frac{\pi}{2} - \theta\right) = \phi'$ for a trajectory $\phi(T)$ near the boundary in the holographic limit.

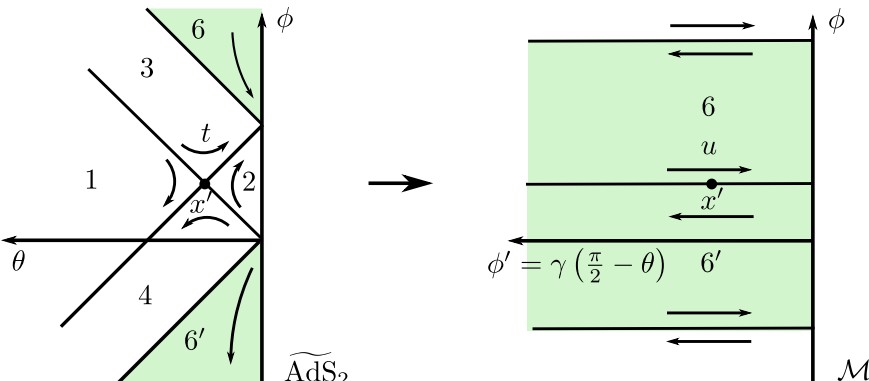

Figure 2: In the asymptotic near-boundary geometry of $\widetilde{\mathrm{AdS}}_2$ given by (15) with $\gamma \to \infty$, light cones are flattened so that only relative regions 6 and its copies remain. In particular, the long-time dynamics in $\widetilde{\mathrm{AdS}}_2$ corresponding to regions 6 and $6'$ extends to vanishing times, or arbitrarily close to reference point $x'$.

the geometry of two and three points on $\mathcal{M}$ with some precision. In the following we relegate the derivation of various geometric results to Appendix A.

**Geometry of two points:** Let us first note the relative coordinates associated with a pair of points $(x_1; x_2)$. An appropriate coordinate for measuring geodesic distance is given by

$$y_{12} = \frac{2\sqrt{\phi_1' \phi_2'}}{\left| \sin\left( \frac{\phi_1 - \phi_2}{2} \right) \right|} .$$ 
(16)

It relates to the bare proper distance between the points on $\widetilde{\mathrm{AdS}}_2$ as $2\gamma / \sqrt{z_{12}} = y_{12} + O\left(\gamma^{-2}\right)$, $z_{12} \sim$ (geodesic distance)$^2$. The remaining relative coordinate measures the direction from which $x_1$ approaches $x_2$, which shifts under isometries fixing $x_2$. Formally, it parameterizes orbits of points under the isotropy group of the (left) action of $\widetilde{\mathrm{SL}}(2, \mathbb{R})$ on $\mathcal{M}$.[12] It is given by

$$u_{12} = \cot\left( \frac{\phi_1 - \phi_2}{2} \right) \phi_2' \mp \frac{\sqrt{\phi_1' \phi_2'}}{\sin\left( \frac{\phi_1 - \phi_2}{2} \right)} ,$$ 
(17)

with upper sign (lower sign) for $n'$ even (odd), $2\pi n' - \pi < \phi_1 - \phi_2 < 2\pi n' + \pi$. See Figure 3a for a depiction of its level curves.

Note that fixing the reference point $x_2$, only the relative region 6 and its copies (defined in [15]) remain in the asymptotic geometry $\mathcal{M}$, demarcated by $2\pi n < \phi_1 - \phi_2 < 2\pi(n+1)$, $n \in \mathbb{Z}$. For our purposes of evaluating integrals in the generator equation (4) in the semiclassical limit, only the relative region 6 ($n = 0$) and region $6'$ ($n = -1$) come into play. In particular, we are interested in short time scales and the limit of target points approaching each other, in which the upper sign in (17) applies. Finally, we note the integration measure over $\mathcal{M}$ can be expressed using relative coordinates $(y, u)$ with respect to some reference point as $d\mathrm{vol}_{\mathcal{M}} = \gamma 16 \, dy \, y^{-3} du$.

**Dynamics in Schwarzian regime:** We recall from [15] that an $\widetilde{\mathrm{SL}}(2, \mathbb{R})$-invariant two-point function for a spin-$\nu$ particle on $\widetilde{\mathrm{AdS}}_2$ takes the general form $\Psi^\nu(x; x') = \left| \frac{\varphi_{23}}{\varphi_{14}} \right|^\nu f_j(z)$, with a

---

[12]This isotropy group is generated by $\Lambda_0 - \Lambda_2$, whereas that for $\widetilde{\mathrm{AdS}}_2$ is generated by $\Lambda_2$.

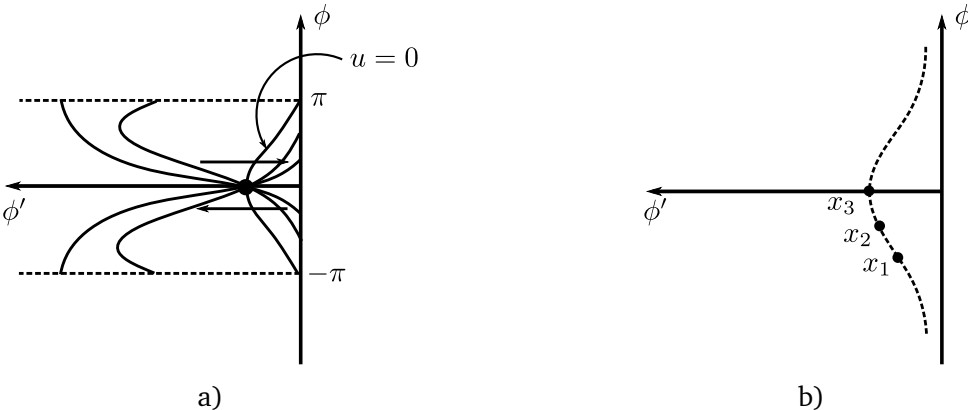

Figure 3: Depiction of a) level curves of relative coordinate $u$ with respect to reference point, and b) points on the classical trajectory of a spin-$\nu$ particle, ordered in increasing proper time.

spin prefactor multiplying a function of geodesic distance, indexed by relative region.[13]

The spin prefactor becomes trivial at short distances in the limit of flat space. However, it is non-trivial in curved space, and we will be interested in the gauge-invariant product of spin factors appearing in the three-event quantum distribution as

$$q_{T_3,T_2,T_1}(x_3,x_2,x_1) \sim \Psi^\nu(x_3;x_1)\Psi^\nu(x_1;x_2)\Psi^\nu(x_2;x_3) \sim e^{\nu\vartheta}, \qquad (18)$$

expressed in the holographic limit $\gamma \to \infty$ and on the asymptotic geometry $\mathcal{M}$. (Note that in the case of the two-event quantum distribution, spin prefactors cancel.) In an integral involving the three-event quantum distribution, this three-point spin-loop factor amounts to an additional large phase as compared to the flat case, which yields in the semi-classical limit a saddle-point configuration of three points that is appropriately curved.

Setting aside the spin prefactor, the radial part of two-point functions appearing in joint quantum distributions of the boundary particle (9) as

$$\langle x_1|e^{-iHT}\rho|x_2\rangle = \int dE\, \frac{e^{-\beta E-iET}}{Z}\mathrm{P}_E(x_1;x_2), \quad \langle x_1|e^{iHT}|x_2\rangle = \int dE\, e^{iET}\,\mathrm{I}_E(x_1;x_2), \quad (19)$$

have the following holographic expansions in the Schwarzian regime:

$$\mathring{\mathrm{P}}_E(x;0) = \gamma^{-1}\frac{\sinh 2\pi s}{2\pi^2}\begin{cases} yK_{2is}(-iy)\left(1+O(\gamma^{-2})\right), & \text{in region } 6', \\ yK_{2is}(iy)\left(1+O(\gamma^{-2})\right), & \text{in region } 6, \end{cases} \qquad (20)$$

$$\mathring{\mathrm{I}}_E(x;0) = \gamma^{-1}\frac{1}{4\pi}\begin{cases} iyI_{2is}(iy)\left(1+O(\gamma^{-2})\right), & \text{in region } 6', \\ -iyI_{-2is}(-iy)\left(1+O(\gamma^{-2})\right), & \text{in region } 6, \end{cases} \qquad (21)$$

where $y$ is the invariant distance in (16). That is, the leading expressions in large $\gamma$ found in [15,16] are unchanged to subleading order.[14] See Appendix B for the derivation. In order to evaluate the generator equation (4) to leading non-vanishing order in large $\gamma$, we need to expand the integrand of each integral to subleading order, which requires the increased accuracy in (20), (21).

---

[13]The spin prefactor is gauge-dependent and here we have shown it in the tilde gauge [15]. Its expression involves the coordinates $\varphi_1 = \phi - \theta + \frac{\pi}{2}$, $\varphi_2 = \phi + \theta - \frac{\pi}{2}$, $\varphi_3 = \phi' - \theta' + \frac{\pi}{2}$, and $\varphi_4 = \phi' + \theta' - \frac{\pi}{2}$, and $\varphi_{kl} = 2\sin\frac{\varphi_k-\varphi_l}{2}$.

[14]In (20), we have factored out $e^{-2\pi\gamma}$ relative to the expression given in [15], since it is eliminated in the denominator $Z$ of the density matrix $\rho$ after renormalization in the holographic limit.

**Geometry of three points:** Let us consider two invariant[15] quantities involving three points $(x_1, x_2, x_3)$ that enter the three-event quantum distribution of a boundary particle on $\mathcal{M}$: the invariant distance $y_{31}$, and the three-point spin-loop phase

$$\omega = 2\left(\phi_1' \frac{\phi_{23}}{\phi_{12}\phi_{31}} + \phi_2' \frac{\phi_{31}}{\phi_{12}\phi_{23}} + \phi_3' \frac{\phi_{12}}{\phi_{31}\phi_{23}}\right), \qquad \phi_{ij} = 2\sin\frac{\phi_i - \phi_j}{2}, \qquad (22)$$

which relates to the spin-loop phase in $\widetilde{\text{AdS}}_2$ defined in (18) as $\gamma\vartheta = \omega + O(\gamma^{-2})$.

In the semi-classical limit the integrals in the generator equation (4) localize near the classical trajectory of the boundary particle which goes up in proper time, that is to the relative regions $(x_1; x_2), (x_2; x_3) \in$ region 6' and $(x_3; x_1) \in$ region 6, with $x_2$ and $x_1$ approaching $x_3$. See Figure 3b. As such it is useful to obtain composition formulas for $y_{31}, \omega$ in terms of relative coordinates $y_{12}, y_{23}$ and $u_{12}, u_{32}$ that apply in the case that $\phi_{32}, \phi_{21}$ are small and positive. Using inverted distance coordinates

$$l_{ij} = y_{ij}^{-1}, \qquad (23)$$

we have

$$l_{31} = l_{12} + l_{23} + 2l_{12}l_{23}(u_{32} - u_{12}), \qquad (24)$$

$$\omega = u_{32} - u_{12} + \frac{l_{12} + l_{23}}{2l_{12}l_{23}} + \frac{l_{12}^2 + l_{23}^2}{2l_{12}l_{23}} \frac{1}{(l_{12} + l_{23} + 2l_{12}l_{23}(u_{32} - u_{12}))}. \qquad (25)$$

We will also make use of the composition formula for $u_{31}$,

$$u_{31} = u_{21} + \frac{l_{23}(u_{32} - u_{12})}{(l_{12} + l_{23} + 2l_{12}l_{23}(u_{32} - u_{12}))}. \qquad (26)$$

## 3.2 Reconstruction of Einstein's equations

As explained in the introduction, in order for a quantum stochastic process to possess a classical limit consisting of a stochastic process with positive joint probability distributions, in addition to its quantum observable having possible values that are dense in a target space, it is important that its joint quantum distributions have a semi-classical limit so that their integrals can be evaluated by the saddle-point method. Furthermore, we have proposed that if the quantum stochastic process is Markovian or locally Markovian in its classical limit, we may expect a quantum generator equation to characterize the entire process, just as a generator equation involving the conditional probability kernel characterizes a classical Markov process.

In our quantum system consisting of the boundary degrees of freedom of JT gravity, the semi-classical limit consists of taking the energy of the boundary particle (12) to be large, that is

$$s^2 \gg 1. \qquad (27)$$

Then the evaluation of integrals involving joint quantum distributions, including those in the generator equation, proceeds by a double expansion: first we take the holographic limit (14), expanding in large $\gamma$, then we take the semi-classical limit (27), expanding in large $s$ with $s/\gamma$ small but finite (corresponding to the bare inverse temperature $L$ being large but finite).[16]

---

[15]With respect to either left or right action of $\widetilde{\text{SL}}(2, \mathbb{R})$.

[16]Since $s/\gamma$ is finite we also express the second expansion as one in large $\gamma$, but it is important to distinguish the two expansions, e.g. we will see the scaling of geometric variables change between the two expansions.

In the semi-classical limit, we find that modified Bessel functions appearing in the two-point functions (20), (21) have an expansion with leading exponential form,

$$\begin{Bmatrix} K_{2is}(\pm il^{-1}) \\ \mp i\pi I_{\mp 2is}(\mp il^{-1}) \end{Bmatrix} = \sqrt{\frac{\pi}{2\sqrt{l^{-2}+4s^2}}} \exp\left(\mp\frac{\pi}{4}i \mp i\sqrt{l^{-2}+4s^2} \pm 2is\sinh^{-1}(2sl)\right)$$

$$\times\left(1 \pm \frac{i}{24}\frac{3l^{-2}-8s^2}{(l^{-2}+4s^2)^{3/2}} + O\left(\gamma^{-2}\right)\right). \tag{28}$$

See Appendix B. Using this expansion, we can perform saddle-point evaluations of integrals involving joint quantum distributions. Let us first note that the joint probability distributions extracted in this way in the classical limit via (2) do not exhibit the Markov property, unlike in the case of JT gravity in flat space. Roughly, this is the statement that with the two-event probability distribution being given by a Gaussian packet, $\int \mathcal{D}x_2\mathcal{D}x_1 q_{T_2,T_1}(x_2,x_1) \approx \int \mathcal{D}l_{21} f_{T_{21}}(l_{21})$, $f_T(l) = \frac{1}{16l_*}\sqrt{\frac{\alpha}{2\pi}}e^{-\frac{1}{2}\alpha(l-l_*)^2}$—$\alpha(\beta,T)$ is a positive constant and $l_*(\beta,T)$ the saddle-point displacement—the three-event probability distribution fails to factorize into a product of Gaussian packets, $\int \mathcal{D}x_3\mathcal{D}x_2\mathcal{D}x_1 q_{T_3,T_2,T_1}(x_3,x_2,x_1) \napprox \int \mathcal{D}l_{32}\mathcal{D}l_{21} f_{T_{32}}(l_{32})f_{T_{21}}(l_{21})$, due to finite-temperature effects in the quantum distribution $q_{T_3,T_2,T_1}$. (Finite-temperature or infrared cutoff effects are invisible in the local limit in which we zoom in near a point and spacetime is flat, see [6]).

However, with the intuition that the quantum stochastic process in curved spacetime still maintains a locally Markovian quality as the spacetime manifold can be sewn together from flat patches, we proceed to evaluate the generator equation (4) involving its conditional quantum distributions, that is, its two- and three-event quantum distributions with single-event quantum distribution (coinciding with probability distribution) factored out. In order to evaluate the equation to leading non-vanishing order in large $\gamma$, we find that the integrand of each integral needs to be twice-expanded to $O(\gamma^{-1})$ accuracy (in the holographic, then semi-classical limit), after which the saddle-point expansion needs to be employed up to sub-subleading order in large $\gamma$. The computation becomes feasible with the help of a closed formula for the saddle-point expansion of a multivariate integral derived in [6] and reproduced in (D.7)-(D.10).

Before outlining the calculation, let us put Einstein's equations of JT gravity shown in (5) in the form that it will be recovered from the generator equation. That is, we write the metric and ensuing covariant derivatives at a point $x_2$ using coordinates $l(x_2;x_1)$, $u(x_2;x_1)$ relative to a point $x_1$ approaching $x_2$. Details of the following derivation are provided in Appendix C.

Switching from absolute coordinates $X^\mu = (\phi_2, \phi_2')$ to the relative coordinates $(l,u)$, we are interested in the asymptotics of the metric (15) as $x_1$ approaches $x_2$ at distances short compared to the AdS radius, that is $\gamma l \sim$ (bare proper distance) $\ll 1$. Expanding in small $l \ll \gamma l \ll 1$, we confirm the leading behavior of the metric is flat, $ds^2 \approx 16(-\gamma^2 dl^2 + l^2 du^2)$, with corrections suppressed by $\gamma^{-1}$ and $\gamma l$. To retain the curvature of the metric as $\gamma l \to 0$, we should include corrections to the leading behavior suppressed by $\gamma l$ and $\gamma^2 l^2$. Assuming $x_1$ approaches $x_2$ from below,

$$ds^2 \approx 16\left(-\gamma^2 dl^2 + l^2\left(1 - 4\gamma^2 l^2\right)du^2 + 4\gamma^2 l^2 dl du\right), \tag{29}$$

using which we obtain that at leading order in $\gamma$,

$$\lim_{l\to 0}\frac{\partial X^\mu}{\partial l}\frac{\partial X^\nu}{\partial l}\left(\nabla_\mu\nabla_\nu - g_{\mu\nu}\nabla^2 - g_{\mu\nu}\Lambda\right)\Phi(X) \approx \lim_{l\to 0}\gamma^2\left(\frac{1}{l^2}\partial_u^2 - \frac{2}{l}\partial_u - 16\right)\Phi(l,u). \tag{30}$$

Since the tensorial components $\frac{\partial X^\mu}{\partial l}\frac{\partial X^\nu}{\partial l}$ have independent dependences on $u$ which measures the direction of $x_2$ relative to the approaching $x_1$,[17] for (30) to vanish regardless of the direction of approach, each tensorial component of Einstein's equations should hold. We will

---

[17]For example, $\frac{\partial \phi_2}{\partial l} = 4\phi_2'$, $\lim_{l\to 0}\frac{\partial \phi_2'}{\partial l} = 4\phi_2'u$.

indeed find that the generator equation (4) reduces to the right-hand-side of (30) being zero, with the constrained probability density $\Phi(x)$ in the quantum problem identified with the area of compactified space $\Phi(x)$ solved for by Einstein's equations!

We give a complete account of the evaluation of the generator equation (4) in Appendix D. Here we outline some key steps involved, using the three-integral on the right-hand-side

$$\int \mathcal{D}x_1 \mathcal{D}x_2 \frac{q_{T_3,T_2,T_1}(x_3,x_2,x_1)}{q_{T_1}(x_1)}\Phi(x_1)$$
$$= \int \mathcal{D}x_1 \mathcal{D}x_2 \frac{\langle x_3|e^{-iHT_{31}}\rho|x_1\rangle\langle x_1|e^{iHT_{21}}|x_2\rangle\langle x_2|e^{iHT_{32}}|x_3\rangle}{\langle x_1|\rho|x_1\rangle}\Phi(x_1). \tag{31}$$

Let us first clarify the fixing of "gauge" or the directional coordinate $u$ appearing in (30). The expectation value $\langle x_1|\rho|x_1\rangle$ is infinite; it can be calculated as

$$\mathbf{1} = \int \mathcal{D}x_1 \operatorname{Tr}(\rho\,|x_1\rangle\langle x_1|) = \int \mathcal{D}x_1 \frac{\langle x_1|\rho|x_1\rangle}{\operatorname{vol}(\widetilde{\mathrm{SL}}(2,\mathbb{R}))} \quad \Rightarrow \quad \langle x_1|\rho|x_1\rangle = \operatorname{vol}(K(x_1)), \tag{32}$$

where $K(x_1)$ is the isotropy group fixing $x_1$ of the left action of $\widetilde{\mathrm{SL}}(2,\mathbb{R})$ on $\mathcal{M}$. Thus the effect of division by $q_{T_1}(x_1)$ inside an integral is to fix a directional coordinate $u$ defined relative to the point $x_1$ at $T_1$—recall the discussion leading to (17). We fix the gauge invariantly across both sides of (4), by fixing the coordinate $u$ of the point at $T_2$ relative to the point at $T_1$. That is, inside the three-point integral (31) we set $\langle x_1|\rho|x_1\rangle^{-1} = \delta(u_{21} - u)$ and inside the two-point integrals, $\langle x_1|\rho|x_1\rangle^{-1} = \delta(u_{31} - u)$, for some constant $u$.

Next, let us consider the holographic expansion of the integrand in (31). Taking into account (20), (21) and the spin-loop phase (22), then restricting to our region of interest $(x_3; x_2), (x_2; x_1) \in$ region 6 to which the integral will localize in the semi-classical limit, the integral becomes

$$\int 16dl_{12}du_{12} \int 16dl_{23}du_{23}\,\delta(u_{21}-u) \int ds\,s\frac{e^{-(\beta+iT_{31})s^2/2}}{Z}\frac{\sinh 2\pi s}{2\pi^2}l_{31}^{-1}K_{2is}\left(il_{31}^{-1}\right)$$
$$\times \int ds_1\,s_1 e^{iT_{21}s_1^2/2}\frac{i}{4\pi}I_{2is_1}\left(il_{12}^{-1}\right) \int ds_2\,s_2 e^{iT_{32}s_2^2/2}\frac{i}{4\pi}I_{2is_2}\left(il_{23}^{-1}\right)e^{-i\omega}\Phi(l_{31},u_{31})\left(1+O\left(\gamma^{-2}\right)\right). \tag{33}$$

Note we have specified the position $x_1$ of $\Phi$ with relative coordinates $(l_{31},u_{31})$, the point $x_3$ being fixed. Anticipating that in the semi-classical limit we will be able to use decomposition formulas (24), (25), (26) for $l_{31}, u_{31}$, and $\omega$, we change variables of integration as $(l_{12},u_{12},l_{23},u_{23}) \to (l_{12},u_{21},l_{23},u_{32}-u_{12})$ which introduces a Jacobian

$$\frac{\partial(l_{12},u_{12},l_{23},u_{23})}{\partial(l_{12},u_{21},l_{23},u_{32}-u_{12})} = \frac{\phi_3'}{\phi_1'}\frac{1}{\cos(\phi_1-\phi_3)} = 1-4u_{31}l_{31}+O\left(l^2\right). \tag{34}$$

The Jacobian is trivial in flat space, with the $O(l)$ correction coming from curvature of space-time. Holding $\gamma l \sim$ (bare proper distance) fixed, $l \sim \frac{\gamma l}{\gamma} \sim O(\gamma^{-1})$, and we only need to retain $O(\gamma^{-1})$ corrections in order to obtain the generator equation to leading non-vanishing order. This formalizes our intuition that curvature and Einstein's equations are determined at second-order displacement from a point, or one order of displacement beyond the flat approximation. Curvature effects at $O(\gamma^{-1})$ also enter the integrand of (33) in the following way. We adapt the Taylor expansion of $\Phi(x_1)$ appearing in (4) to relative coordinates, and expand $\Phi(l_{31},u_{31})$ about $l_{31} = l_{21}, u_{31} = u_{21}$. This involves solving for $\Delta x_1 = x_1 - \bar{x}_1$ such that $l_{3\bar{1}} = l_{21}$ and

$u_{3\bar{1}} = u_{21}$, and subsequently the displacement of relative coordinates

$$\Delta l_{31} = l_{13} - l_{12} + 2(l_{13} - l_{12})(l_{13}u_{31} - l_{12}u_{21}) + O(l^3),$$

$$\Delta u_{31} = \frac{l_{13}}{l_{12}}(u_{31} - u_{21}) + \frac{1}{l_{12}}\left((l_{13}u_{31} - l_{12}u_{21})(l_{12}u_{21} - 2l_{13}u_{21} + l_{13}u_{31}) + (l_{13} - l_{12})^2 \phi_3'^2\right)$$
$$+ O(l^2). \tag{35}$$

The subleading terms in $l$ are again curvature corrections to flat spacetime, and result in $O(\gamma^{-1})$ terms in the Taylor expansion

$$\Phi(l_{31}, u_{31}) = \Phi(l_{21}, u_{21}) + \Delta l_{31}\Phi^{(1,0)}(l_{21}, u_{21}) + \Delta u_{31}\Phi^{(0,1)}(l_{21}, u_{21})$$
$$+ \frac{1}{2}\Delta l_{31}^2 \Phi^{(2,0)}(l_{21}, u_{21}) + \Delta l_{31}\Delta u_{31}\Phi^{(1,1)}(l_{21}, u_{21}) + \frac{1}{2}\Delta u_{31}^2 \Phi^{(0,2)}(l_{21}, u_{21}) + \cdots \tag{36}$$

Finally, we take the semi-classical limit, using the expansion (28) for two-point functions and composition formulas (24), (25), (26), and evaluating the integral by expansion about its saddle-point,

$$s_* = s_1^* = s_2^* = \frac{2\pi}{\beta}, \qquad l_{12}^* = \frac{1}{2s_*}\sinh\left(\frac{s_* T_{21}}{2}\right), \quad l_{23}^* = \frac{1}{2s_*}\sinh\left(\frac{s_* T_{32}}{2}\right),$$
$$u_{32}^* - u_{12}^* = s_*\left(\tanh\left(\frac{s_* T_{21}}{2}\right) + \tanh\left(\frac{s_* T_{32}}{2}\right)\right). \tag{37}$$

With $s/\gamma$ being small but finite, that is $s \sim O(\gamma)$ in the semi-classical expansion, we note that coordinates $u_{32} - u_{12}$, $u_{21}$ which were $O(1)$ in the holographic expansion recover their bare scaling $O(\gamma)$.[18] Having expanded to $O(\gamma^{-1})$ in the holographic limit, we again expand to $O(\gamma^{-1})$ in the semiclassical limit, taking this new scaling into account. The result is that terms remaining in the three-integral (31) after cancellation by those in the two-integrals of (4) give

$$0 = \left[\lim_{x' \to x} -\frac{i}{32}\left(\frac{1}{l^2}\partial_u^2 - \frac{2}{l}\partial_u - 16\right)\Phi(l, u)\right]_{(x;x') \in \text{region 6}}, \tag{38}$$

with $u$ fixed and arbitrary, where we have relabeled $x_3 = x$, and $l = l(x; x')$, $u = u(x; x')$.[19] But from (30), these are nothing but Einstein's equations of JT gravity with negative cosmological constant, with probability density $\Phi(x)$ identified as the dilaton field or area of compactified space in the gravity theory!

Before further discussion in the next section, let us take stock of the physical content of the generator equation (4) employing joint quantum distributions given by EVPP's (1). Emphatically, it is not solving for the *actual* probability distribution of the observable $X$ in state $\rho$. This is given by the single-event distribution $q_T(dx) = \mathcal{D}x \, \text{Tr}\left(\rho e^{iHT} P(x) e^{-iHT}\right)$; in our system we have a thermal state and the corresponding probability density $q_T(x) = q_T(dx)/\mathcal{D}x$, $q_T(x) = q(x) = \text{Tr}(\rho |x\rangle\langle x|)$ is uniform and vanishingly small, a natural consequence of the quantum state respecting the isometry of the target space. (This is to be contrasted with the probability density $\Phi(x)$ solving the generator equation, which is finite and has a non-trivial profile.) Neither is the generator equation redundant with the Schrodinger equation expressing the time evolution of the quantum state; if it were, in our case it would simply express that

---

[18] That is, their scaling with respect to coordinates in $\widetilde{\text{AdS}}_2$ rather than the asymptotic geometry $\mathcal{M}$.

[19] Terms of order three or higher in the Taylor expansion (36) do not contribute to this result at leading non-vanishing order in $\gamma$, producing terms of $O(T_{32}^2)$. However, they can contribute at higher orders in the large $\gamma$ expansion, because the saddle-point expansion strips away increasing powers of $l_{23}$, or $T_{32}$.

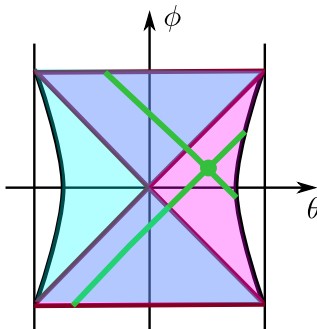

Figure 4: Semi-classically, the probability at a point of the black hole solution can only flow to inside of light-cones (shown in green). Thus in the two-sided black hole solution, the effective area at a point outside the left (right) horizon consists of probability for the left (right) boundary particle to be there, whereas the effective area inside the horizon is probability for either the left or right particle to be there.

a thermal quantum state does not evolve in time, $[H, \rho] = 0$. Rather, the generator equation is the quantum generalization of an evolution equation involving conditional probabilities, and asks the question, "*Assuming* some probability measure exists, is it compatible with evolution according to *conditional* quantum distributions associated with the dynamics of the quantum observable?"—its solutions are probability measures answering in the affirmative.

We can apply this understanding that the generator equation is solving for consistency—rather than existence—of probability,[20] to the black hole solution in anti-de Sitter JT gravity. Recall that we derived a generator equation (hitherto known as Einstein's equations) from the dynamics of a single boundary particle in a thermal state. Yet we know the black hole solution satisfying the equation is associated with two boundary particles. This is possible as the generator equation does not specify which, if any, particles are contributing to the probability at a given point; it is simply solving for a probability measure, which if it existed, would be consistent with the measure itself evolving according to thermal dynamics. In the case of the black hole solution, we know that the probability in the solution is attributable to two distinct particles in a finely-tuned entangled state where each is in a thermal state [19]. Furthermore, we can recognize that there is a qualitative distinction between the dilaton field—or effective spacetime area—inside and outside the black hole horizon, when understood as probability density. The effective area at a point outside the horizon consists of probability for a single particle—the boundary particle outside the horizon—to be there, whereas the area at a point inside the horizon consists of probability for *either* of the two boundary particles to be there.[21] See Figure 4.

As emphasized in [6], it is in fact not necessary for a quantum stochastic process to be defined by EVPP's of a quantum system—its mathematical definition only requires a sequence of joint quantum distributions which each sum to one and satisfy marginalization relations with respect to each other—nor is a probability solution to the generator equation of a process with locally Markovian classical limit required to be attributable to a configuration of actual quantum systems. This allows our framework to extend to quantum gravity beyond the anti-de Sitter setting, in which a gravitational theory is associated with quantum systems residing at the time-like boundary of anti-de Sitter space. In [6] we saw this mechanism at work in the case of flat JT gravity, where the (Markovian) quantum stochastic process giving rise to a generator equation was one consisting of joint quantum distributions obtained by taking an

---

[20]Relatedly, recall that we are solving for probability measures which do not have to integrate to 1.

[21]This is true as along as we are in the semi-classical limit in which the flow of particle probability is confined to inside of light-cones.

asymptotic limit of those of a boundary quantum system of anti-de Sitter JT gravity—and thus divorced from the dynamics of the said quantum system.

## 4 Conclusion

Let us try to further enunciate the physical significance of a generator equation as we have defined it, in particular by making a precise comparison between the generator equation (4) using EVPP's (1) of a quantum system, and the Schrodinger equation expressing the time evolution of the density matrix of the quantum system, $\frac{d}{dT}\rho(T) = -i[H, \rho(T)]$. We will be using notation applicable to the more general case in which a sub-measure $\mathcal{D}x$ does not simultaneously factor out of joint quantum distributions $q_{T_n,\dots T_1}(dx_n, \dots dx_1)$ and the probability measure $\nu(dx)$.

The left-hand-side of the generator equation involves the time derivative of the two-event quantum distribution,

$$\partial_{T_2} q_{T_2, T_1}(dx_3, dx_1) = \text{Tr}\Big(\rho e^{iHT_1} P(dx_1) e^{-iH(T_1-T_2)} i[H, P(dx_3)] e^{-iHT_2}\Big). \tag{39}$$

Recall that in probability-theoretic language, we are representing events of a quantum system with projection operators, e.g. the event that "observable $X$ assumes value in the range $(x_1, x_1+dx_1)$ at time $T_1$" with the projection operator $e^{iHT_1} P(dx_1) e^{-iHT_1}$ in the Heisenberg picture.[22] (As mentioned in the Introduction, we are not concerned with making measurements on the quantum state which would involve its collapse, but rather studying joint *quantum* distributions associated with logical conjunctions of prescribed events at different times, in the coherent and unmeasured quantum state $\rho$.[23]) Thus (39) divided by $q_{T_1}(dx_1)$ expresses the time evolution of the event that "X assumes value in $(x_3, x_3 + dx_3)$ at time $T_2$" *conditioned* on the event that "X assumes value in $(x_1, x_1 + dx_1)$ at time $T_1$" while in the quantum state $\rho$, or more precisely, the time-derivative of the corresponding conditional quantum distribution, (3). This kernel then acts upon the assumed probability measure for $X$, $\nu(dx_1)$.

On the right-hand-side of the generator equation, we take the time-derivative *after* acting with the conditional quantum distribution on the measure; we have $\partial_{T_2}(QM_{T_2,T_1} \nu)(dx_3) = \partial_{T_2} \int_{x_1} \kappa_{T_2,T_1}(dx_3; x_1) \nu(dx_1)$.[24] Mathematically, the equation is non-trivial because differentiation and integration do not commute when the integrand is not continuous in the relevant region, and this is the case at hand as $\lim_{T_{21} \to 0} \kappa_{T_2,T_1}(dx_3, x_1) \propto \delta(x_3 - x_1)$. Physically, the equation is expressing a consistency condition for some probability measure $\nu$ to be evolving with respect to conditional quantum distribution $\kappa_{T_2,T_1}$, and can be understood as the quantum equation of motion for time evolution of the latter, encapsulated by (39), transferred to the measure in the instantaneous limit $T_2 \to T_1$. (In JT gravity, it is necessary to express the derivative on the right-hand-side as we have done in (4)—inserting an intermediate point at time $T_2$ using the marginalization relation $q_{T_3,T_1}(dx_3, dx_1) = \int_{x_2} q_{T_3,T_2,T_1}(dx_3, dx_2, dx_1)$—in order to factor out $\langle x_1|\rho|x_1\rangle \sim q_{T_1}(dx_1)$ consistently between the two sides of the equation as a coordinate's worth of volume, where the coordinate is of the point at $T_2$ relative to that at $T_1$.)

Thus once the quantum stochastic process at hand satisfies the first-order condition of its observable having possible values that are dense in target space, so that a continuum approximation is possible, the equation holds as a formal constraint on probability measures evolving

---

[22]See Appendix A of [6] for further elaboration.

[23]In contrast to the classical case, the logical conjunction of events is ordered and not necessarily an event—the product of projectors $P_1 \cdots P_n$ is a projector only if the projectors $P_1, \cdots, P_n$ commute amongst themselves. This is the reason we cannot associate a joint probability distribution, only a joint quantum distribution.

[24]We are denoting by *QM* the quantum Markov operator that maps measures forward in time with kernel $\kappa$.

with respect to it. Yet it is only if the joint quantum distributions of the process have a semi-classical limit and we work in that limit, that the equation can probe the local "shape" of joint quantum distributions, resulting in a local constraint. (Of course, independently of whether the equation is local or non-local, it may not possess any solutions at all.) Furthermore, it is only in the case of the process being Markovian or locally Markovian, that the equation can be expected to have any relation to characterizing or reconstructing the quantum stochastic process at finite times, in particular its higher-order joint quantum distributions.[25]

We have reached the overwhelming conclusion that from the point of view of quantum theory, the volume measure of spacetime should be understood as a probability measure evolving and thus constrained with respect to some quantum stochastic process.

Let us comment on the relation between our work and known facets of AdS/CFT, including the Ryu-Takayanagi formula which served as a motivation for our work. Note in deriving our results we did not make any use of AdS/CFT relations. The latter are UV/IR relations between quantities in a microscopic theory and an effective gravity theory at low energies. In contrast our starting point for constructing the gravity theory—that spacetime is the target space of a quantum observable, and the volume measure is a probability measure for it—were already relations at the level of the low-energy theory. As such, in their current form, they are best understood not as "new entries in the holographic dictionary", but as relations we have conjectured should hold generally between spacetime and corresponding quantum degrees of freedom, in a bottom-up sense quantized theory of gravity. It is an interesting problem to try to "lift" our results to the UV-IR setup of AdS/CFT duality; we are trying to address this in current work in progress.

We conclude with some further questions and open problems.

One that is important in terms of completing our theory is a precise characterization of classical stochastic processes that are locally Markovian, and thus of quantum stochastic processes having a locally Markov classical limit. More generally, we would like to have a good understanding of the precise conditions under which a quantum stochastic process gives rise to gravity and spacetime.

We would also like to be able to investigate examples in which the quantum generator/gravitational equations are non-linear, and to arrive at a direct probabilistic interpretation of the metric and gravitational action. Relatedly and perhaps as a prior step, one could try to incorporate matter excitations on top of the thermal state in our analysis, and attempt to understand the signifiance of the energy-momentum tensor in relation to spacetime being composed of probability.

Another direction of improvement would be to formulate the quantum stochastic process in AdS JT gravity that we have studied using microscopic correlators in the Sachdev-Ye-Kitaev (SYK) model [12, 20–22]. One can imagine, for example, of being able to compute stringy corrections to Einstein's equations in the gravity theory corresponding to the SYK model, after understanding how fermionic correlators of the model naturally "latch onto" our generator equation involving correlators in the quantum theory of the boundary of AdS JT gravity. In this respect, it is useful that we have found Einstein's equations can be recovered entirely from the Schwarzian regime, as this is the dynamical regime in which one has analytic control of correlators in the SYK model.

Finally, but not of least conceptual interest, is the problem of deriving the Ryu-Takayanagi formula from our identification of the volume measure of spacetime as a probability measure. It is clear that at minimum this would involve understanding, say in the context of AdS JT gravity, a rationale for normalizing the probability density/dilaton function with respect to the density matrix of the boundary quantum system.

---

[25]Intuitively, we are equating this reconstruction of the stochastic process with obtaining an entire spacetime (probability) manifold from a local equation.

## Acknowledgments

We thank David Gross, Juan Maldacena, and Edward Witten for discussions, and Mark Sred-nicki for encouragement and suggestions on presentation.

**Funding information**   This work was supported by the Simons Foundation through the "It from Qubit" Simons Collaboration, by a grant to the KITP from the Simons Foundation (216179, LB), and in part by the National Science Foundation under Grant No. NSF PHY-1748958. It was completed at the Aspen Center for Physics, which is supported by National Science Foundation grant PHY-1607611.

## A  Asymptotic geometry near boundary of $\widetilde{\text{AdS}}_2$

The asymptotic geometry $\mathcal{M}$ near a boundary of $\widetilde{\text{AdS}}_2$ has the metric (15), with time coordinate $\phi$ inherited from global $\widetilde{\text{AdS}}_2$ and radial coordinate $\phi' = \gamma(\frac{\pi}{2} \mp \theta)$ near the boundary $\theta = \pm\frac{\pi}{2}$. Without loss of generality, we work with the geometry near the right boundary $\theta = \frac{\pi}{2}$.

Besides global coordinates $(\phi, \phi')$, an alternative set of coordinates which will be of use to us are the Poincaré coordinates

$$q = \tan\frac{\phi}{2}, \qquad q' = \frac{\phi'}{2\cos^2\frac{\phi}{2}}. \tag{A.1}$$

These parametrize the asymptotic near-boundary geometry of a Poincaré patch of global $\widetilde{\text{AdS}}_2$, see Figure 5a.

### A.1  Geometry of two points $(x_1, x_2)$

The relative coordinate $y_{12}$ given in (16), which measures geodesic distance, can be derived by expressing the distance coordinate $z_{12} = \varphi_{13}\varphi_{24}/\varphi_{12}\varphi_{34}$ in $\widetilde{\text{AdS}}_2$—where $\varphi_1 = \phi_1 - \theta_1 + \frac{\pi}{2}$, $\varphi_2 = \phi_1 + \theta_1 - \frac{\pi}{2}$ and $\varphi_3 = \phi_2 - \theta_2 + \frac{\pi}{2}$, $\varphi_4 = \phi_2 + \theta_2 - \frac{\pi}{2}$ are coordinates of $x_1$ and $x_2$, respectively, and $\varphi_{ij} = 2\sin\frac{\varphi_i - \varphi_j}{2}$—in terms of global coordinates on $\mathcal{M}$,

$$\varphi_1 = \phi_1 + \frac{\phi_1'}{\gamma}, \quad \varphi_2 = \phi_1 - \frac{\phi_1'}{\gamma}, \quad \varphi_3 = \phi_2 + \frac{\phi_2'}{\gamma}, \quad \varphi_4 = \phi_2 - \frac{\phi_2'}{\gamma}, \tag{A.2}$$

then expanding in large $\gamma$:

$$\boxed{\frac{2\gamma}{\sqrt{z_{12}}} = y_{12} + O\left(\gamma^{-2}\right), \qquad y_{12} = \frac{2\sqrt{\phi_1'\phi_2'}}{\left|\sin\left(\frac{\phi_1 - \phi_2}{2}\right)\right|}.} \tag{A.3}$$

The relative coordinate $u_{12}$ given in (17) measures the direction of $x_1$ relative to $x_2$. Let us first recall that the relativistic light-cone in $\widetilde{\text{AdS}}_2$ flattens in $\mathcal{M}$ as in Figure 2, after which only the relative regions $6^{(n)}$, $n \in \mathbb{Z}$ remain, demarcated by $2\pi n < \phi_1 - \phi_2 < 2\pi(n+1)$. Then we observe that $u_{12}$, being an analogue in the deformed geometry of the relative Schwarzschild time $t_{12}$ in $\widetilde{\text{AdS}}_2$, will be spacelike in each region. To derive the expression (17), we proceed as follows.

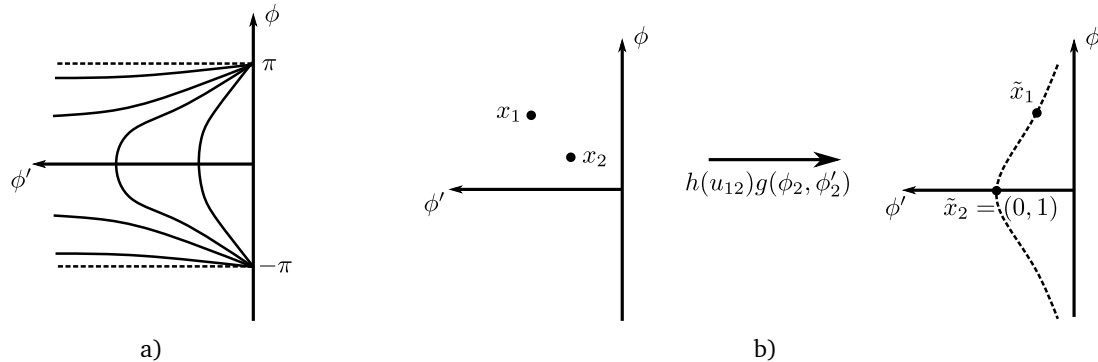

Figure 5: Depiction of a) a Poincare patch of the asymptotic space $\mathcal{M}$, and level curves of its spatial coordinate $q'$, and b) the map fixing $u_{12}$, which maps $x_1$ to a point with same spatial Poincaré coordinate as the base point $(\phi = 0, \phi' = 1)$.

The group $\widetilde{\mathrm{SL}}(2, \mathbb{R})$ can be parametrized with three "Euler angles" [23]

$$g(\xi, \varphi, \vartheta) = e^{\varphi \Lambda_0} e^{\xi \Lambda_1} e^{-\vartheta \Lambda_0}, \qquad \xi \geq 0. \tag{A.4}$$

We can represent elements of the group as fractional maps on the complex plane,

$$\begin{pmatrix} a & b \\ c & d \end{pmatrix} z = \frac{az + b}{cz + d}, \qquad \forall z \in \mathbb{C}, \tag{A.5}$$

where the generators of the group are given by the fractional maps

$$\Lambda_0 = \frac{1}{2} \begin{pmatrix} i & 0 \\ 0 & -i \end{pmatrix}, \qquad \Lambda_1 = \frac{1}{2} \begin{pmatrix} 0 & 1 \\ 1 & 0 \end{pmatrix}, \qquad \Lambda_2 = \frac{1}{2} \begin{pmatrix} 0 & i \\ -i & 0 \end{pmatrix}. \tag{A.6}$$

Then the space $\mathrm{AdS}_2$ consists of the product of two unit circles, $z_1 = e^{i\varphi_1}, z_2 = e^{i\varphi_2}$ where the phases $\varphi_1, \varphi_2$ relate to global coordinates on $\widetilde{\mathrm{AdS}}_2$ as $\varphi_1 = \phi - \theta + \frac{\pi}{2}, \varphi_2 = \phi + \theta - \frac{\pi}{2}$. We work with the space $\mathcal{M}$ by embedding it in $\widetilde{\mathrm{AdS}}_2$,

$$\varphi_1 = \phi + \frac{\phi'}{\gamma}, \qquad \varphi_2 = \phi - \frac{\phi'}{\gamma}, \tag{A.7}$$

and keeping only first-order corrections in large $\gamma$. Then we find the isotropy subgroup of $\widetilde{\mathrm{SL}}(2, \mathbb{R})$ fixing the base point $(\phi = 0, \phi' = 1)$ is parametrized as

$$h(u) = e^{u(\Lambda_0 - \Lambda_2)}, \tag{A.8}$$

while the group element

$$g(\phi, \phi') = e^{\ln \phi' \Lambda_1} e^{-\phi \Lambda_0}, \tag{A.9}$$

takes an arbitrary point $(\phi, \phi')$ to the base point. Consequently, the isotropy subgroup fixing an arbitrary point $x_2$ is given by

$$K(x_2) = \{ g(\phi_2, \phi_2')^{-1} h(u) g(\phi_2, \phi_2') | u \in \mathbb{R} \}. \tag{A.10}$$

Now, given $(x_1; x_2)$, we fix $u_{12}$ by requiring that the transformation $h(u_{12}) g(\phi_2, \phi_2')$—which maps $x_2$ to the base point, $\widetilde{x}_2 = (\widetilde{\phi}_2 = 0, \widetilde{\phi}_2' = 1)$—maps $x_1$ to a point with same spatial Poincaré coordinate $q'$ (see (A.1)) as the base point, that is

$\widetilde{\phi}_1'/\left(2\cos^2\frac{\widetilde{\phi}_1}{2}\right)=\widetilde{\phi}_2'/\left(2\cos^2\frac{\widetilde{\phi}_2}{2}\right)=\frac{1}{2}$. See Figure 7. Also using that $y_{12}$ must be preserved by the transformation, $2\sqrt{\phi_1'\phi_2'}/\left|\sin\left(\frac{\phi_1-\phi_2}{2}\right)\right|=2\sqrt{\widetilde{\phi}_1'}/\left|\sin\frac{\widetilde{\phi}_1}{2}\right|$, we solve for $\tilde{\phi}_1, \widetilde{\phi}_1'$ and find

$$u_{12}=\cot\left(\frac{\phi_1-\phi_2}{2}\right)\phi_2'\mp\frac{\sqrt{\phi_1'\phi_2'}}{\sin\left(\frac{\phi_1-\phi_2}{2}\right)}\,. \tag{A.11}$$

The correct branch cuts that preserve relative regions are given by

$$2\pi n'-\pi<\phi_1-\phi_2<2\pi n'+\pi\,,\quad n'\begin{cases}\text{even: upper sign,}\\\text{odd: lower sign.}\end{cases} \tag{A.12}$$

Note that there is a branch cut at the center of each relative region $6^{(n)}$; this is consistent with the deformation from $\widetilde{\text{AdS}}_2$ to $\mathcal{M}$, see Figure 2. The first term in (A.11) corresponds to the directional relative coordinate $t_{12}=\frac{1}{2}\ln\left(\left|\varphi_{14}\varphi_{24}/\varphi_{13}\varphi_{23}\right|\right)$ in $\widetilde{\text{AdS}}_2$ which parameterizes orbits of points under the isotropy group $H(x_2)$ of the action of $\widetilde{\text{SL}}(2,\mathbb{R})$ on $\widetilde{\text{AdS}}_2$ (cf. (A.10)):

$$\gamma t_{12}=\cot\left(\frac{\phi_1-\phi_2}{2}\right)\phi_2'+O(\gamma^{-2})\,. \tag{A.13}$$

From (A.3) and (A.13), we see that the bare scaling of relative coordinates $y_{12}, u_{12}$, that is, their scaling with respect to coordinates on $\widetilde{\text{AdS}}_2$ as opposed to those on the asymptotic geometry $\mathcal{M}$, are given by $y_{12}, u_{12}\sim O(\gamma)$.

## A.2   Geometry of three points $(x_1, x_2, x_3)$

Let us derive composition formulas for the invariant distance

$$y_{13}=y_{31}=\frac{2\sqrt{\phi_1'\phi_3'}}{\left|\sin\left(\frac{\phi_1-\phi_3}{2}\right)\right|}\,, \tag{A.14}$$

and three-point spin-loop phase (22) that were given in (24), (25). The formulas can be checked directly using the definitions of variables $l_{12}, l_{23}$, and $u_{12}, u_{32}$. But to derive them in the first place, we proceed as follows.

We fix $x_2$ to be the base point, $x_2=(\phi_2=0, \phi_2'=1)$, and invert the formulas

$$\begin{aligned}y_{12}&=\frac{2\sqrt{\phi_1'}}{\left|\sin\frac{\phi_1}{2}\right|}\,, & u_{12}&=\cot\left(\frac{\phi_1}{2}\right)\mp\frac{\sqrt{\phi_1'}}{\left|\sin\frac{\phi_1}{2}\right|}\,,\\ y_{23}&=\frac{2\sqrt{\phi_3'}}{\left|\sin\frac{\phi_3}{2}\right|}\,, & u_{32}&=\cot\left(\frac{\phi_3}{2}\right)\mp\frac{\sqrt{\phi_3'}}{\left|\sin\frac{\phi_3}{2}\right|}\,,\end{aligned} \tag{A.15}$$

to solve for $\phi_1, \phi_1', \phi_3, \phi_3'$. We choose branch cuts corresponding to configurations of three points that are relevant in the semi-classical limit at small proper times, that is, $\phi_{21}, \phi_{32}$ small and positive (negative), applying to the case of the quantum particle having spin $\nu$ ($-\nu$).[26] These are

$$\begin{aligned}\sqrt{\phi_1'}&=\frac{y_{12}}{2\sqrt{1+(u_{12}\mp\frac{y_{12}}{2})^2}}\,, & \mp\sin\left(\frac{\phi_1}{2}\right)&=\frac{1}{\sqrt{1+(u_{12}\mp\frac{y_{12}}{2})^2}}\,,\\ \sqrt{\phi_3'}&=\frac{y_{32}}{2\sqrt{1+(u_{32}\pm\frac{y_{32}}{2})^2}}\,, & \pm\sin\left(\frac{\phi_3}{2}\right)&=\frac{1}{\sqrt{1+(u_{32}\pm\frac{y_{32}}{2})^2}}\,,\end{aligned} \tag{A.16}$$

---

[26]In particular, since $\phi_{21}, \phi_{32}$ are small in magnitude, we choose the upper sign in formulas for $u_{12}, u_{32}$ in (A.15), which apply when $-\pi<\phi_1-\phi_2, \phi_2-\phi_3<\pi$.

with upper (lower) signs corresponding to $\phi_{32}, \phi_{21}$ small and positive (negative). Plugging into (A.14) and (22), we obtain the general formulas

$$y_{31} = \frac{1}{2} \frac{y_{12} y_{23}}{(\pm(u_{32} - u_{12}) + (y_{12} + y_{23})/2)}, \tag{A.17}$$

$$\pm\omega = \pm(u_{32} - u_{12}) + \frac{y_{12} + y_{23}}{2} + \frac{y_{12}^2 + y_{23}^2}{4} \frac{1}{(\pm(u_{32} - u_{12}) + (y_{12} + y_{23})/2)}. \tag{A.18}$$

Meanwhile, composition formulas for

$$u_{31} = \cot\left(\frac{\phi_3 - \phi_1}{2}\right)\phi_1' - \frac{\sqrt{\phi_3' \phi_1'}}{\sin\left(\frac{\phi_3 - \phi_1}{2}\right)}, \tag{A.19}$$

can be obtained in the same way,

$$u_{31} = u_{21} + \frac{y_{12}(u_{32} - u_{12})}{2(\pm(u_{32} - u_{12}) + (y_{12} + y_{23})/2)}. \tag{A.20}$$

Note all composition formulas are invariant under the left-action of $\widetilde{\mathrm{SL}}(2, \mathbb{R})$ fixing $x_2$ or $x_1$, which manifest as translations in $u_{32}, u_{12}$ and $u_{31}, u_{21}$, respectively.

# B    Two-point functions in Schwarzian regime

Here we study two-point functions of the boundary particle of JT gravity in the Schwarzian regime, or in the holographic limit and at long time scales. In the holographic limit,

$$\gamma \gg 1, \qquad s^2 \ll \gamma^2, \tag{B.1}$$

the particle likes to be near the boundary of $\widetilde{\mathrm{AdS}}_2$, i.e. its single-particle wavefunctions are localized there. Furthermore, over long time scales,

$$T_b \gg 1, \tag{B.2}$$

it stays in the near-boundary region, i.e. its two-point functions also localize to near the boundary. Thus in the Schwarzian regime, we are reduced to studying the dynamics of the particle in the asymptotic geometry near the boundary of $\widetilde{\mathrm{AdS}}_2$, the spacetime $\mathcal{M}$ defined in (15).

## B.1    Exact two-point functions in Schwarzian regime

In order to obtain two-point functions in the Schwarzian regime, we take the limit $\gamma \to \infty$ of the exact two-point functions of JT gravity found in [15], in region 5, 6 and their copies—these are the regions remaining in the asymptotic geometry, the asymptotic space near the right (left) boundary of $\widetilde{\mathrm{AdS}}_2$ dividing into regions $6^{(n)}$ ($5^{(n)}$) after fixing a reference point (see Figure 2). In the following we will be showing formulas in regions $5, 6', 5', 6$, to which two-point functions localize in the semi-classical limit.[27]

---

[27]These are the results we will be using in our calculation of the generator equation of the quantum stochastic process of the particle. However, formulas in other copies of regions 5, 6 can be found in a similar manner.

As can be found in Sections 4.2 and 5.2 of [15], the radial part of two-point functions corresponding to the density operator P and propagator I are given by

$$\mathring{P}_E(x;0) = \rho(E) \begin{cases} \Gamma(\lambda+\nu)\Gamma(1-\lambda+\nu)A_{\lambda,\nu,\nu}(w^{-1}), & \text{in region } 5,6', \\ \Gamma(\lambda-\nu)\Gamma(1-\lambda-\nu)A_{\lambda,-\nu,-\nu}(w^{-1}), & \text{in region } 5',6, \end{cases} \tag{B.3}$$

$$\mathring{I}_E(x;0) = \frac{1}{(2\pi)^2} \begin{cases} \Gamma(\lambda+\nu)\Gamma(1-\lambda+\nu)A_{\lambda,\nu,\nu}(w^{-1}) \\ \quad +\Gamma(\lambda-\nu)\Gamma(1-\lambda-\nu)A_{\lambda,-\nu,-\nu}(w^{-1})e^{2\pi i\lambda}, & \text{in region } 5,6', \\ \Gamma(\lambda+\nu)\Gamma(1-\lambda+\nu)A_{\lambda,\nu,\nu}(w^{-1})e^{2\pi i\lambda} \\ \quad +\Gamma(\lambda-\nu)\Gamma(1-\lambda-\nu)A_{\lambda,-\nu,-\nu}(w^{-1}), & \text{in region } 5',6, \end{cases} \tag{B.4}$$

where

$$\rho(E) = (2\pi)^{-2}\frac{\sinh(2\pi s)}{(\cosh(2\pi\gamma)+\cosh(2\pi s))}, \tag{B.5}$$

is the exact density of states in JT gravity, $\lambda = \frac{1}{2}+is$ is the energy parameter (cf. (12)) and $\nu = \mp i\gamma$ the spin of the particle, and

$$w = \frac{z}{z-1}, \tag{B.6}$$

is an invariant distance coordinate on $\widetilde{\mathrm{AdS}}_2$.

In order to expand in large $\gamma$, it is useful to switch from the basis of functions

$$A_{\lambda,\pm\nu,\pm\nu}(w^{-1}) = (w^{-1})^{\pm\nu}(1-w^{-1})^\lambda \mathbf{F}(\lambda\pm\nu,\lambda\pm\nu,1\pm2\nu;w^{-1}), \tag{B.7}$$

where $\mathbf{F}(a,b,c;z)$ is the regularized hypergeometric function, to the basis

$$B_{\lambda,\nu,\nu}(w^{-1}) = (w^{-1})^{\pm\nu}(1-w^{-1})^\lambda \mathbf{F}(\lambda\pm\nu,\lambda\pm\nu,2\lambda;1-w^{-1}),$$
$$B_{1-\lambda,\nu,\nu}(w^{-1}) = (w^{-1})^{\pm\nu}(1-w^{-1})^{1-\lambda}\mathbf{F}(1-\lambda\pm\nu,1-\lambda\pm\nu,2(1-\lambda);1-w^{-1}), \tag{B.8}$$

so that $\nu$ only appears in the first two arguments of hypergeometric functions:

$$\Gamma(\lambda+\nu)\Gamma(1-\lambda+\nu)A_{\lambda,\nu,\nu}(w^{-1})$$
$$= \frac{\pi}{\sin 2\pi\lambda}\left(\frac{\Gamma(\lambda+\nu)}{\Gamma(1-\lambda+\nu)}B_{\lambda,\nu,\nu}(w^{-1}) - \frac{\Gamma(1-\lambda+\nu)}{\Gamma(\lambda+\nu)}B_{1-\lambda,\nu,\nu}(w^{-1})\right). \tag{B.9}$$

Then we use the series representation

$$\left(\frac{1}{2}x\right)^{c-1}\mathbf{F}\left(A+\frac{c}{2},A+\frac{c}{2},c;\frac{x^2}{4A^2}\right) = I_{c-1}(x)\left(1+\frac{1}{A}\left(\frac{x}{2}\right)^2\right)+O\left(\frac{1}{A^2}\right), \tag{B.10}$$

which converges for $\left|\frac{x^2}{4A^2}\right| < 1$, and which can be obtained by expanding each term in the hypergeometric series on the left-hand-side; $I_\alpha(x)$ is the modified Bessel function of the first kind.

Without loss of generality, let us fix $\nu = -i\gamma$. Applying (B.10) to (B.8) and also noting the large-$\gamma$ expansion

$$\frac{\Gamma(\lambda+\nu)}{\Gamma(1-\lambda+\nu)} = e^{\pi s}\gamma^{2is}\left(1+O(\gamma^{-2})\right), \tag{B.11}$$

and that of $2\gamma/\sqrt{z}$ in (A.3), we have

$$\frac{\Gamma(\lambda+\nu)}{\Gamma(1-\lambda+\nu)}B_{\lambda,\nu,\nu}(w^{-1}) = \frac{1}{2\gamma}\left(\frac{2\gamma}{\sqrt{z}}\right)e^{2\pi s}I_{2is}\left(i\frac{2\gamma}{\sqrt{z}}\right)\left(1+O\left(\gamma^{-2}\right)\right)$$
$$= \frac{y}{2\gamma}e^{2\pi s}I_{2is}(iy)\left(1+O\left(\gamma^{-2}\right)\right). \tag{B.12}$$

Finally, using the modified Bessel function of the second kind $K_\alpha(x) = \frac{\pi}{2\sin\pi\alpha}(I_{-\alpha}(x) - I_\alpha(x))$,

$$\Gamma(\lambda \pm \nu)\Gamma(1 - \lambda \pm \nu)A_{\lambda,\pm\nu,\pm\nu}(w^{-1}) = \frac{y}{\gamma}K_{2is}(\mp iy)\left(1 + O\left(\gamma^{-2}\right)\right). \tag{B.13}$$

In particular, we find the Schwarzian propagators found in [15,16] are unchanged to subleading order in large $\gamma$,

$$\mathring{P}_E(x;0) = e^{-2\pi\gamma}\gamma^{-1}\frac{\sinh 2\pi s}{2\pi^2}\begin{cases} yK_{2is}(-iy)\left(1 + O\left(\gamma^{-2}\right)\right), & \text{in region } 5,6', \\ yK_{2is}(iy)\left(1 + O\left(\gamma^{-2}\right)\right), & \text{in region } 5'6, \end{cases} \tag{B.14}$$

$$\mathring{I}_E(x;0) = \gamma^{-1}\frac{1}{4\pi}\begin{cases} iyI_{2is}(iy)\left(1 + O(\gamma^{-2})\right), & \text{in region } 5,6', \\ -iyI_{-2is}(-iy)\left(1 + O\left(\gamma^{-2}\right)\right), & \text{in region } 5',6. \end{cases} \tag{B.15}$$

### B.2 Exponential form in semi-classical limit

We can obtain an exponential form for the two-point functions (B.14), (B.15) in the semi-classical limit

$$s^2 \gg 1, \tag{B.16}$$

by evaluating the integral

$$K_{2is}(\pm iy) = \frac{1}{2}\int_{-\infty\pm\frac{\pi}{2}i}^{\infty\mp\frac{\pi}{2}i} d\xi\, e^{-p(\xi)}, \qquad p(\xi) = \pm iy\cosh\xi \mp 2is\xi, \tag{B.17}$$

using the saddle-point method. There is a saddle-point at

$$\sinh\xi_* = \frac{2s}{y}, \tag{B.18}$$

where

$$p(\xi_*) = \pm i\sqrt{y^2 + 4s^2} \mp 2is\sinh^{-1}\left(\frac{2s}{y}\right), \qquad p''(\xi_*) = \pm i\sqrt{y^2 + 4s^2}. \tag{B.19}$$

Thus we deform the integration contour in (B.17) as in Figure 6. Using the closed formula for the saddle-point expansion of an integral of a single variable, and also taking the large-$s$ limit of the analytic continuation

$$\mp i\pi I_{\mp 2is}(\mp iy) = K_{2is}(\pm iy) - e^{-2\pi s}K_{2is}(\mp iy), \tag{B.20}$$

we obtain the result in (28).

## C Einstein's equations in relative coordinates

Here we give the derivation of the metric and Einstein's equations (29), (30) at a point of the asymptotic geometry $\mathcal{M}$, written using relative coordinates with respect to an approaching point.

At a point $x_2$, we switch from using absolute coordinates $X^\mu = (\phi_2, \phi_2')$ to relative coordinates $l = l_{21}$, $u = u_{21}$ with respect to an approaching point $x_1$. With $\phi_{21}$ being small,

$$l = \frac{\left|\sin\left(\frac{\phi_2 - \phi_1}{2}\right)\right|}{2\sqrt{\phi_2'\phi_1'}}, \qquad u = \cot\left(\frac{\phi_2 - \phi_1}{2}\right)\phi_1' - \frac{\sqrt{\phi_2'\phi_1'}}{\sin\left(\frac{\phi_2 - \phi_1}{2}\right)}, \tag{C.1}$$

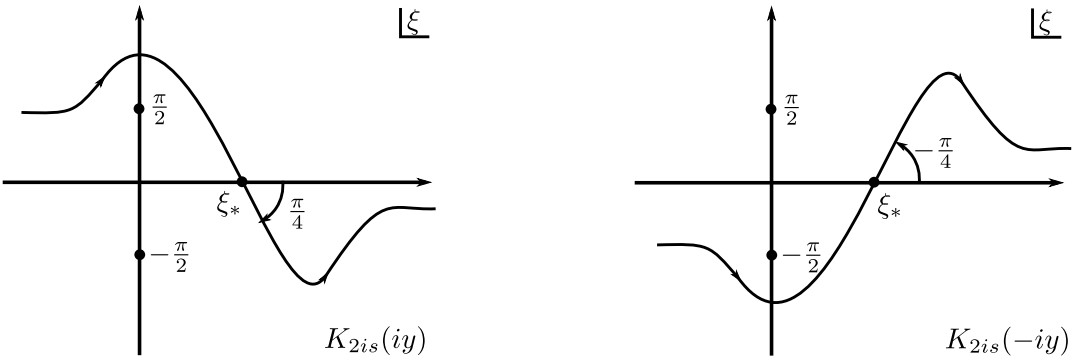

Figure 6: Deformation of contours in (B.17) to pass through saddle-points.

and solving for derivatives of $X^\mu$ with respect to $(l,u)$,

$$\frac{\partial \phi_2}{\partial l} = \pm 4\phi_2', \qquad \frac{\partial \phi_2}{\partial u} = -\frac{\sin^2\left(\frac{\phi_2-\phi_1}{2}\right)}{\phi_1'},$$

$$\frac{\partial \phi_2'}{\partial l} = \pm 4\phi_2'\left(\phi_2'\cot\left(\frac{\phi_2-\phi_1}{2}\right) - \frac{\sqrt{\phi_2'\phi_1'}}{\sin\left(\frac{\phi_2-\phi_1}{2}\right)}\right), \qquad \frac{\partial \phi_2'}{\partial u} = -\frac{\phi_2'}{\phi_1'}\sin(\phi_2-\phi_1),$$

$$(C.2)$$

with the upper (lower) sign for $\phi_{21}$ positive (negative). Inverting (C.1) to solve for $\sin\left(\frac{\phi_1-\phi_2}{2}\right)$, $\frac{\phi_1'}{\phi_2'}$ in terms of $l,u$ at fixed $x_2$,

$$\sin\left(\frac{\phi_1-\phi_2}{2}\right) = \mp\sqrt{\frac{1-\sqrt{1-16l^2(1\pm 2ul)^2\phi_2'^2}}{2}}, \qquad \frac{\phi_1'}{\phi_2'} = \frac{1-\sqrt{1-16l^2(1\pm 2ul)^2\phi_2'^2}}{8l^2\phi_2'^2},$$

$$(C.3)$$

and substituting in (C.2), we can obtain the metric (15) at $x_2$ using coordinates $(l,u)$.

We consider its asymptotics as $x_1$ approaches $x_2$ at distances short compared to the AdS radius, $\gamma l \ll 1$. Expanding in small $l \ll \gamma l \ll 1$,

$$g_{uu} \approx 16l^2 + O\left(l^3\right) + \underbrace{O\left(\gamma^2 l^4\right)}_{-64\gamma^2 l^4},$$

$$g_{ll} \approx -16\gamma^2 + O(1), \qquad g_{ul} \approx O(l) + \underbrace{O\left(\gamma^2 l^2\right)}_{\pm 32\gamma^2 l^2},$$

$$(C.4)$$

and we confirm the leading behavior of the metric is flat, with corrections suppressed by $\gamma^{-1}$ and $\gamma l$. The curvature of the metric can be retained by including corrections suppressed by $\gamma l$ and $\gamma^2 l^2$ which we have shown explicitly in (C.4),

$$\boxed{ds^2 \approx 16\left(-\gamma^2 dl^2 + l^2\left(1-4\gamma^2 l^2\right)du^2 \pm 4\gamma^2 l^2 dl\,du\right).}$$

$$(C.5)$$

The resulting Laplacian, at leading order in $\gamma$, is

$$\nabla^2 = g^{\mu\nu}\nabla_\mu\nabla_\nu \approx \frac{1}{16}\left(\frac{1}{l^2}\partial_u^2 \pm \frac{2}{l}\partial_u\right).$$

$$(C.6)$$

Also noting

$$\frac{\partial X^\mu}{\partial l}\frac{\partial X^\nu}{\partial l}g_{\mu\nu} = g_{ll} \approx -16\gamma^2, \qquad \frac{\partial X^\mu}{\partial l}\frac{\partial X^\nu}{\partial l}\nabla_\mu\nabla_\nu = \nabla_l\nabla_l \approx \mp 4\gamma^2\frac{1}{l}\partial_u, \qquad (C.7)$$

we obtain

$$\lim_{l \to 0} \frac{\partial X^\mu}{\partial l} \frac{\partial X^\nu}{\partial l} \left( \nabla_\mu \nabla_\nu - g_{\mu\nu} \nabla^2 - g_{\mu\nu} \Lambda \right) \approx \lim_{l \to 0} \gamma^2 \left( \frac{1}{l^2} \partial_u^2 \mp \frac{2}{l} \partial_u - 16 \right), \qquad \text{(C.8)}$$

whose tensorial components are differential operators appearing in Einstein's equations of JT gravity. The expression can be verified by computing the left-hand-side in absolute coordinates $(\phi_2, \phi_2')$, and comparing with the right-hand-side computed using the short-distance asymptotics of the derivatives in (C.2),

$$\frac{\partial \phi_2}{\partial l} = \pm 4\phi_2', \qquad \frac{\partial \phi_2}{\partial u} \approx -8\phi_2' l^2,$$
$$\frac{\partial \phi_2'}{\partial l} \approx \mp 4\phi_2' u + 8\phi_2' \left( u^2 - \phi_2'^2 \right) l, \qquad \frac{\partial \phi_2'}{\partial u} \approx \mp 4\phi_2' l + 8\phi_2' u l^2. \qquad \text{(C.9)}$$

Note $\partial_l, l^{-1} \partial_u$ are the scaled derivatives finite as $l \to 0$, and their behavior is odd as $x_1$ approaches $x_2$ from below and above.

# D  Evaluation of generator equation

Here we give a complete account of the evaluation of the generator equation (4) for the quantum stochastic process of the boundary AdS JT gravity in the Schwarzian regime, which leads to Einstein's equations in the final form (38). Without loss of generality, we consider the case of the boundary particle having spin $\nu = -i\gamma$, and its quantum motion on the asymptotic geometry near the right boundary of $\widetilde{\text{AdS}}_2$.

## D.1  Two-event integrals

Let us consider the two-event integral appearing on the right-hand-side of the equation,

$$\int \mathcal{D}x_1 \frac{q_{T_2,T_1}(x_3,x_1)}{q_{T_1}(x_1)} \Phi(x_1) = \int \mathcal{D}x_1 \frac{\langle x_3 | e^{-iHT_{21}} \rho | x_1 \rangle \langle x_1 | e^{iHT_{21}} | x_3 \rangle}{\langle x_1 | \rho | x_1 \rangle} \Phi(x_1). \qquad \text{(D.1)}$$

Noting the factor $q_{T_1}(x_1)^{-1}$ fixes the coordinate $u_{31} = u$ as explained near (32) of Section 3.2, and using expansions (20), (21) of two-point functions (19), we have that in the holographic limit the integral restricted to the relative region $(x_3; x_1) \in$ region 6 becomes[28]

$$\int 16 \, dl \, du_{13} \, \delta(u_{31} - u) \int ds \, s \frac{e^{-(\beta + iT)s^2/2}}{Z} \frac{\sinh 2\pi s}{2\pi^2} l^{-1} K_{2is} \left( i l^{-1} \right)$$
$$\times \int ds' \, s' e^{iT s'^2/2} \frac{i}{4\pi} I_{2is'} \left( i l^{-1} \right) \Phi(l, u_{31}) \left( 1 + O\left( \gamma^{-2} \right) \right), \qquad \text{(D.2)}$$

where we have used the shorthand $T_{21} = T$, $l_{13} = l_{31} = l$. Changing variable in the integral from $u_{13}$ to $u_{31}$ introduces the Jacobian

$$\frac{du_{13}}{du_{31}} = \frac{\phi_3'}{\phi_1'} \frac{1}{\cos(\phi_1 - \phi_3)} = 1 - 4u_{31} l + O(l^2). \qquad \text{(D.3)}$$

---

[28] After taking the semi-classical limit, the integral will localize to this region—i.e. there will be no saddle-point in the other region $(x_3; x_1) \in$ region 6'—as the classical trajectory of a spin-$\nu$ particle goes up near the right boundary of $\widetilde{\text{AdS}}_2$.

As explained near (34), we need—and actually should—only retain $O(l) \sim O(\gamma^{-1})$ corrections for our purposes of calculating the generator equation to leading non-vanishing order.[29]

Next, we take the semi-classical limit using the expansion (28), which gives

$$\frac{1}{2\pi^3 Z} \int dl \int ds \int ds' \, e^{-p(l,s,s')} \frac{ss'}{l(l^{-2}+4s^2)^{1/4}(l^{-2}+4s'^2)^{1/4}} \Phi(l,u)$$
$$\times (1-4ul) \left( 1 + \frac{i}{24} \left( \frac{3l^{-2}-8s^2}{(l^{-2}+4s^2)^{3/2}} - \frac{3l^{-2}-8s'^2}{(l^{-2}+4s'^2)^{3/2}} \right) + O(\gamma^{-2}) \right), \tag{D.4}$$

$$p(l,s,s') = (\beta + iT)\frac{s^2}{2} - 2\pi s - \frac{iTs'^2}{2} + i\left( \sqrt{l^{-2}+4s^2} - 2s \sinh^{-1} 2sl \right)$$
$$- i\left( \sqrt{l^{-2}+4s'^2} - 2s' \sinh^{-1} 2s'l \right). \tag{D.5}$$

We find there is a saddle point at

$$\boxed{s_* = s'_* = \frac{2\pi}{\beta}, \qquad l^*_{31} = \frac{\sinh\left(\frac{s_* T_{21}}{2}\right)}{2s_*} = \frac{\beta}{4\pi} \sinh\left(\frac{\pi T_{21}}{\beta}\right),} \tag{D.6}$$

consistent with the classical trajectory of the boundary particle in the Schwarzian regime, see e.g. Appendix C of [6].

To evaluate the integral we use the closed-form formula for the saddle-point expansion of a multi-variable integral derived in [6]. If there are $\mathcal{N}$ variables $\boldsymbol{z} = (z_1, \ldots, z_\mathcal{N})$,

$$\int d\boldsymbol{z} \, f(\boldsymbol{z}) e^{-p(\boldsymbol{z})} = e^{-p_*} \sum_{m=0}^{\infty} \sum_{j=0}^{2m} \sum_{|\boldsymbol{i}|=3j}^{2(m+j)} \frac{(-1)^j}{j!} \hat{B}_{ij}(p)$$
$$\times \sum_{|\boldsymbol{k}|=2(m+j)-|\boldsymbol{i}|} f_{\boldsymbol{k}} \prod_{\mathcal{M}=1}^{\mathcal{N}} \frac{1}{2}\left(1+(-1)^{(\boldsymbol{k}+\boldsymbol{i})_\mathcal{M}}\right) p_\mathcal{M}^{-((\boldsymbol{k}+\boldsymbol{i})_\mathcal{M}+1)/2} \Gamma\left(\frac{(\boldsymbol{k}+\boldsymbol{i})_\mathcal{M}+1}{2}\right), \tag{D.7}$$

where $p_\mathcal{M}$ are coefficients of quadratic terms in the expansion of the exponent $p(\boldsymbol{z})$ about the saddle-point, and coefficients of higher-order terms in the expansion have been packaged into a function $p : \mathbb{N}^\mathcal{N} \to \mathbb{C}$,

$$p(\boldsymbol{z}) - p(\boldsymbol{z}_*) = \sum_{\mathcal{M}=1}^{\mathcal{N}} p_\mathcal{M}(z_\mathcal{M} - z_\mathcal{M}^*)^2 + \sum_{|\boldsymbol{k}|=3}^{\infty} p_{\boldsymbol{k}}(\boldsymbol{z} - \boldsymbol{z}_*)^{\boldsymbol{k}}. \tag{D.8}$$

$f_{\boldsymbol{k}}$ are coefficients in the expansion of the amplitude function $f(\boldsymbol{z})$,

$$f(\boldsymbol{z}) = \sum_{|\boldsymbol{k}|=0}^{\infty} f_{\boldsymbol{k}}(\boldsymbol{z} - \boldsymbol{z}_*)^{\boldsymbol{k}}, \tag{D.9}$$

---

[29]For example, as $u \sim O(1)$ in large-$\gamma$ counting in the holographic limit, but $u \sim O(s) \sim O(\gamma)$ in large-$\gamma$ counting after taking the semi-classical limit, individual terms that are $O(u^2l^2) \sim O(\gamma^{-2})$ in holographic counting formally become $O(1)$ in semi-classical counting. However, they are actually scaling as $s^2/\gamma^2$, so should not be included in the evaluation to $O(\gamma^{-1})$, $\gamma \to \infty$. To avoid such anomalous contributions at all orders in the saddle-point expansion in the semi-classical limit, we should not include any $O(\gamma^{-2})$ terms in the holographic expansion.

and $\hat{B}_{ij}(x)$, $x : \mathbb{N}^{\mathcal{N}} \to \mathbb{C}$ are multivariate Bell polynomials [24] which can be computed using recursion relations

$$\hat{B}_{ij}(x) = \begin{cases} \delta_{i,0}, & j = 0, \\ x_i, & j = 1, \\ \displaystyle\sum_{|r|=J(j-1)}^{|i|-J} \hat{B}_{i,j-1}(x)\hat{B}_{i-r,1}(x), & j \geq 2, \end{cases} \tag{D.10}$$

where $J$ is an integer s.t. $x_i = 0$ for $|i| < J$—in our case with $x_i = p_i$, $J = 3$. Note we have used the multivariate notation $|i| = \sum_{\mathcal{M}=1}^{\mathcal{N}} i_{\mathcal{M}}$, $z^i = \prod_{\mathcal{M}=1}^{\mathcal{N}} z_{\mathcal{M}}^{i_{\mathcal{M}}}$ for $i \in \mathbb{N}^{\mathcal{N}}$, and in (D.10) $0$ is the vector of 0's in $\mathbb{N}^{\mathcal{N}}$. Importantly, the formula (D.7) assumes the quadratic expansion of the exponent is diagonal in variables $z$, see (D.8) .

Going back to the integral (D.4), we proceed by identifying second derivatives of the exponent which are non-vanishing at the saddle point. They are

$$a = \left.\frac{\partial^2 p}{\partial s^2}\right|_*, \quad a' = \left.\frac{\partial^2 p}{\partial s'^2}\right|_*, \quad b = \left.\frac{\partial^2 p}{\partial s \partial l}\right|_* = -\left.\frac{\partial^2 p}{\partial s' \partial l}\right|_*, \tag{D.11}$$

so denoting $X = s - s_*$, $X' = s' - s'_*$, $Y = l - l_*$, we can complete the squares in the quadratic expansion of (D.5) as

$$p - p_* \approx \frac{1}{2}a\left(X + \frac{b}{a}Y\right)^2 + \frac{1}{2}a'\left(X' - \frac{b}{a'}Y\right)^2 + \frac{1}{2}\left(-b^2\left(\frac{1}{a} + \frac{1}{a'}\right)\right)Y^2. \tag{D.12}$$

Thus we change variables in the integral as

$$s = w - \frac{b}{a}l, \quad s' = w' + \frac{b}{a'}l, \qquad l, \tag{D.13}$$

when applying (D.7) with $\mathcal{N} = 3$. Note the index $m$ in (D.7) counts inverse powers of large exponent—in our case, $\gamma$—relative to the leading term. We would like to evaluate the integral to sub-leading accuracy in large $\gamma$, so we calculate $m = 0, 1$ terms in (D.7) for leading terms in the amplitude in (D.4), and the $m = 0$ term for subleading terms in the amplitude. As explained near (37), the variable $u$ newly acquires the scaling $O(\gamma)$ in semi-classical counting, so both terms in the factor $1 - 4ul$ of the amplitude contribute at leading order. The result of such evaluation is given by

$$\int \mathcal{D}x_1 \frac{q_{T_2,T_1}(x_3,x_1)}{q_{T_1}(x_1)}\Phi(x_1) = \left(1 - 4ul_{31} + \underbrace{c_{00} + c_{10}\partial_{l_{31}} + c_{20}\partial_{l_{31}}^2}_{O(\gamma^{-1})\text{ terms}} + O(\gamma^{-2})\right)\Phi(l_{31},u)\Bigg|_*, \tag{D.14}$$

where the coefficients

$$c_{00} = \frac{iul_{31}(\pi + i\sinh^{-1}(2sl_{31}))\sinh^{-1}(2sl_{31})}{\pi s}, \dots, \tag{D.15}$$

are not important as corresponding terms (and in fact all terms in (D.14)) will be cancelled by those in the zeroth-order term of the Taylor expansion in the three-event integral in (4), see (D.28).

Performing a similar calculation as above, the two-integral on the left-hand-side of the generator equation evaluates to

$$\lim_{T_{21}\to 0^+} \int \mathcal{D}x_1 \frac{\partial_{T_2}q_{T_2,T_1}(x_3,x_1)}{q_{T_1}(x_1)}\Phi(x_1) = \left[\lim_{x_1\to x_3}\left(\frac{1}{4}\partial_{l_{31}} - u + O\left(\gamma^{-1}\right)\right)\Phi(l_{31},u)\right]_{(x_3;x_1)\in\text{region 6}}. \tag{D.16}$$

Note we only need to compute to $O(\gamma^{-1})$ to match the computation of integrals to $O(\gamma^{-2})$ on right-hand-side, as the latter are divided by $T_{32} = (T_b)_{32}/\gamma \sim O(\gamma^{-1})$. The action of $\partial_{T_2}$ brings down a factor of $E' - E = (s'^2 - s^2)/2$ in (D.4), so the integral is enhanced by a factor of $\gamma^2$ in semi-classical counting. Thus we expand around the saddle-point to order $m = 2$ for leading terms in the amplitude, and to order $m = 1$ for subleading terms—the $m = 0$ evaluation right at the saddle-point is identically zero as $s_* = s'_*$.

## D.2 Three-event integrals

Let us consider the three-event integral in the generator equation (31). As explained in Section 3.2, the Taylor expansion of $\Phi(x_1)$ in (4) will be replaced with the expansion (36) adapted to our use of relative coordinates. As outlined in the same Section and also Section 3.1, there are extra components in the evaluation of (31) as compared to the case of two-event integrals, having to do with i) including the spin-loop phase in the three-event joint quantum distribution, ii) identifying all corrections due to curvature in the holographic limit, and iii) using the geometry of three points in the semi-classical limit.

Similarly as in (D.2), noting the factor $q_{T_1}(x_1)^{-1}$ fixes $u_{21} = u$ (the directional coordinate of the point at time $T_2$ relative to the point at time $T_1$), we have that in the holographic limit the integral restricted to the relative regions $(x_3; x_2), (x_2; x_1) \in$ region 6 becomes (33). We must take care to find all $O(\gamma^{-1})$ corrections in the holographic expansion that are due to the curvature of spacetime; these are identified in (34), (35), and (36). Here let us give the derivation of the displacements (35). Solving for $\bar{x}_1$ such that $l_{3\bar{1}} = l_{21}$ and $u_{3\bar{1}} = u_{21}$—note $x_3$ is fixed—we find (cf. (C.3))

$$\sin\left(\frac{\bar{\phi}_1 - \phi_3}{2}\right) = -\sqrt{\frac{1 - \sqrt{1 - 16l_{12}^2(1 + 2u_{21}l_{12})^2\phi_3'^2}}{2}},$$

$$\frac{\bar{\phi}_1'}{\phi_3'} = \frac{1 - \sqrt{1 - 16l_{12}^2(1 + 2u_{21}l_{12})^2\phi_3'^2}}{8l_{12}^2\phi_3'^2}. \tag{D.17}$$

Employing similar solutions for $x_1$ in terms of $l_{31}$ and $u_{31}$, and converting the displacements $\Delta x_1 = x_1 - \bar{x}_1$ as

$$\Delta x_{31} = \frac{\partial x_{31}}{\partial \phi_1}\bigg|_{x_1 = \bar{x}_1} \Delta\phi_1 + \frac{\partial x_{31}}{\partial \phi_1'}\bigg|_{x_1 = \bar{x}_1} \Delta\phi_1', \qquad x_{31} = (l_{31}, u_{31}), \tag{D.18}$$

we find the expansions in (35).

Next, similarly as in (D.4), we take the semi-classical limit of (33), which gives

$$\frac{1}{Z\pi^4}\sqrt{\frac{2}{\pi}} \int dl_{12} \int dl_{23} d(u_{32} - u_{12}) \int ds\, ds_1\, ds_2\, e^{-p(l_{12},l_{23},u_{32}-u_{12},s,s_1,s_2)}$$

$$\times \frac{ss_1s_2}{l_{13}} \frac{e^{\frac{\pi}{4}i}}{(l_{13}^{-2} + 4s^2)^{1/4}(l_{12}^{-2} + 4s_1^2)^{1/4}(l_{23}^{-2} + 4s_2^2)^{1/4}} F(l_{12}, l_{23}, u, u_{32} - u_{12})$$

$$\times \left(1 + \frac{i}{24}\left(\frac{3l_{13}^{-2} - 8s^2}{(l_{13}^{-2} + 4s^2)^{3/2}} - \frac{3l_{12}^{-2} - 8s_1^2}{(l_{12}^{-2} + 4s_1^2)^{3/2}} - \frac{3l_{23}^{-2} - 8s_2^2}{(l_{23}^{-2} + 4s_2^2)^{3/2}}\right) + O(\gamma^{-2})\right), \tag{D.19}$$

$$p = (\beta + iT_{31})\frac{s^2}{2} - 2\pi s - \frac{iT_{21}s_1^2}{2} - \frac{iT_{32}s_2^2}{2} + i\left(\sqrt{l_{13}^{-2} + 4s^2} - 2s\sinh^{-1} 2sl_{13}\right)$$

$$- i\left(\sqrt{l_{12}^{-2} + 4s_1^2} - 2s_1\sinh^{-1} 2s_1l_{12}\right) - i\left(\sqrt{l_{23}^{-2} + 4s_2^2} - 2s_2\sinh^{-1} 2s_2l_{23}\right) + i\omega, \tag{D.20}$$

where the function $F$ includes terms of up to $O(l) \sim O(\gamma^{-1})$ in the product of the Jacobian (34) and Taylor expansion (36) with $u_{21} = u$, and we use composition formulas for $l_{31}$, $u_{31}$, and $\omega$ given in (24), (26), and (25). There is a saddle point at (37).

Next, we proceed to calculate second derivatives of (D.20) and change variables to diagonalize its quadratic expansion about the saddle. The second derivatives non-vanishing at the saddle-point are

$$a = \frac{\partial^2 p}{\partial s^2}\Big|_*, \quad a_j = \frac{\partial^2 p}{\partial s_j^2}\Big|_*, \quad b_j' = \frac{\partial^2 p}{\partial s \partial l_{j,j+1}}\Big|_*, \quad b_j = \frac{-\partial^2 p}{\partial s_j \partial l_{j,j+1}}\Big|_*, \quad c = \frac{\partial^2 p}{\partial \Delta u^2}\Big|_*,$$

$$d = \frac{\partial^2 p}{\partial s \partial \Delta u}\Big|_*, \quad e_j = \frac{\partial^2 p}{\partial \Delta u \partial l_{j,j+1}}\Big|_*, \quad f_{jk} = \frac{-\partial^2 p}{\partial l_{j,j+1} \partial l_{k,k+1}}\Big|_*, \quad j,k = 1,2,$$
(D.21)

where we have used the shorthand $\Delta u = u_{32} - u_{12}$. Denoting $X = s - s_*$, $X_j = s_j - s_j^*$, $Y_j = l_{j,j+1} - l_{j,j+1}^*$, $Z = \Delta u - \Delta u_*$, we find that we can complete the squares in the quadratic expansion of (D.20) as

$$p - p_* \approx \frac{1}{2}\alpha\left(X + \frac{1}{\alpha}\sum_{j=1}^2 b_j Y_j\right)^2 + \frac{1}{2}\sum_{j=1}^2 a_j\left(X_j - \frac{b_j}{a_j}Y_j\right)^2$$
(D.22)

$$+ \frac{1}{2}\sum_{j=1}^2\left(-b_j^2\left(\frac{1}{\alpha} + \frac{1}{a_j}\right)\right)Y_j^2 - \frac{1}{\alpha}b_1 b_2 Y_1 Y_2 + \frac{1}{2}c\left(Z + \frac{d}{c}X + \sum_{j=1}^2\frac{e_j}{c}Y_j\right)^2, \quad \alpha = a - \frac{d^2}{c}.$$

Thus in applying (D.7) with $\mathcal{N} = 6$, we change variables to

$$w = s + \frac{1}{\alpha}\sum_{j=1}^2 b_j l_{j,j+1}, \quad w_j = s_j - \frac{b_j}{a_j}l_{j,j+1}, \quad u = \Delta u + \frac{d}{c}s + \sum_{j=1}^2\frac{e_j}{c}l_{j,j+1},$$
(D.23)

and also diagonalize the matrix of derivatives w.r.t. $l_{j,j+1}$ by working with rotated variables

$$v_+ = l_{12} - \chi l_{23}, \quad v_- = \chi l_{12} + l_{23},$$
(D.24)

$$\chi = -\frac{\left(\alpha(a_1 b_2^2 - a_2 b_1^2) - a_1 a_2(b_1^2 - b_2^2)\right)}{2a_1 a_2 b_1 b_2}\left(1 - \sqrt{1 + \frac{4a_1^2 a_2^2 b_1^2 b_2^2}{\left(\alpha(a_1 b_2^2 - a_2 b_1^2) - a_1 a_2(b_1^2 - b_2^2)\right)^2}}\right).$$
(D.25)

The eigenvalues of the latter matrix are given by

$$\alpha_{\pm} = \frac{\alpha(a_1 b_2^2 + a_2 b_1^2) + a_1 a_2(b_1^2 + b_2^2) \mp \left(\alpha(a_1 b_2^2 - a_2 b_1^2) - a_1 a_2(b_1^2 - b_2^2)\right)}{2\alpha a_1 a_2}$$

$$\times \sqrt{1 + \frac{4a_1^2 a_2^2 b_1^2 b_2^2}{\left(\alpha(a_1 b_2^2 - a_2 b_1^2) - a_1 a_2(b_1^2 - b_2^2)\right)^2}},$$
(D.26)

and the exponent (D.20) expands quadratically with respect to new variables (D.24) as

$$p - p_* \sim -\frac{1}{2}\left(\frac{1}{1+\chi^2}\right)\left(\alpha_+ V_+^2 + \alpha_- V_-^2\right), \quad V_{\pm} = v_{\pm} - v_{\pm}^*.$$
(D.27)

Then the result of evaluating terms of degree $0, 1, 2$ in the Taylor expansion (36) inside of $F$ in (D.19), and expanding about the saddle point to order $m = 1$ for leading terms in the respective amplitudes and to order $m = 0$ for subleading terms, is as follows:

$$\int \mathcal{D}x_1 \mathcal{D}x_2 \frac{q_{T_3,T_2,T_1}(x_3,x_2,x_1)}{q_{T_1}(x_1)} \Phi(x_{31}=x_{21}) = \Bigg( 1 - 4ul_{21} - uT_{32}$$
$$+ c_{20}\partial_{l_{21}}^2 + c_{10}\partial_{l_{21}} + c_{00} + \frac{i}{2}T_{32} + O\left(T_{32}T_{21}, T_{32}^2\right) + O\left(\gamma^{-2}\right) \Bigg) \Phi(l_{21},u)\Bigg|_* , \quad \text{(D.28)}$$

$$\int \mathcal{D}x_1 \mathcal{D}x_2 \frac{q_{T_3,T_2,T_1}(x_3,x_2,x_1)}{q_{T_1}(x_1)} \sum_{|\boldsymbol{k}|=1} \frac{\Phi^{(\boldsymbol{k})}(x_{31}=x_{21})}{\boldsymbol{k}!}\Delta x_{31}^{\boldsymbol{k}}$$
$$= \left( T_{32}\left(\frac{1}{4}\partial_{l_{21}} + \frac{i}{16}l_{21}^{-1}\partial_u\right) + O\left(T_{32}T_{21}, T_{32}^2\right) + O\left(\gamma^{-2}\right) \right) \Phi(l_{21},u)\Bigg|_* , \quad \text{(D.29)}$$

and

$$\int \mathcal{D}x_1 \mathcal{D}x_2 \frac{q_{T_3,T_2,T_1}(x_3,x_2,x_1)}{q_{T_1}(x_1)} \sum_{|\boldsymbol{k}|=2} \frac{\Phi^{(\boldsymbol{k})}(x_{31}=x_{21})}{\boldsymbol{k}!}\Delta x_{31}^{\boldsymbol{k}}$$
$$= \left( T_{32}\left(-\frac{i}{32}l_{21}^{-2}\partial_u^2\right) + O\left(T_{32}T_{21}, T_{32}^2\right) + O\left(\gamma^{-2}\right) \right) \Phi(l_{21},u)\Bigg|_* . \quad \text{(D.30)}$$

To $O(\gamma^{-1})$, terms of degree 3 or higher in the Taylor expansion (36) only give terms of $O(T_{32}^2)$, so do not contribute to the generator equation (4). Collecting the integrals (D.14), (D.16), (D.28), (D.29), (D.30), we obtain (38), with all surviving terms originating from the three-event integral. Note the term corresponding to cosmological constant comes from (D.28).

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
