# Peer review of "Probabilistic deconstruction of a theory of gravity, Part II: curved space"

_SciPost Physics, doi:SciPost Phys. 16, 012 (2024)_

## Round 2 · Referee Report · Anonymous (Referee 1) · 2023-9-27

Strengths

  1. Novel take on a timely subject
  2. Has potential to be generalized

Weaknesses

  1. Very difficult to read and hard to understand the flow of logic.
  2. Does not give due credit to earlier attempts which have attempted to give answers to the motivational questions in the introduction.

Report

This is a second in the series of two papers by the author. In the first paper, a similar line of questioning was asked and Einsteins equations of JT gravity were almost derived [barring the cosmological constant]. In this paper the author derives the full equations introducing a new notion of "sufficiently Markovian in a local sense" probability distributions which admit a classical saddle.

The following points may help improve presentation and the flow of logic which at times is very hard to follow. I also could not see what the conceptual leap in this paper is compared to the first one (except for Markovian to local Markovian) and hence cannot make up my mind about its suitability in SciPost. Currently, I feel it is better suited for SciPost Core.

  1. In the introduction, second para, the author makes it seem that she is the first one to try and address this issue of the quantum information theoretic interpretation of Einstein's equation. This has already been tackled in numerous papers in the literature and goes by the name of 1st law of entanglement. The author should cite these original papers and attempt explain in what manner her proposal hopes to distinguish and lead to new insights compared to these older works.

  2. End of page 2, it is stated that mostly such kernels will be considered which depend only on time differences. Will this automatically restrict to theories with time translation symmetry? [and hence exclude cosmological scenarios?]

  3. The spectrum of the bulk theory should consider a spin-2 massless particle. How does one think of higher spin theories in the language of probability distributions?

  4. Section 2 is titled Generators of non-Markovian process. I don't see any generators being given in this section explicitly.

  5. While I see -16 in equation 38, I struggled to see what exactly the difference was compared to the first instalment that enabled one to get the cosmological constant.

  6. Apart from the interesting first sentence in the abstract, I did not feel that the connection with Ryu-Takayanagi was clearly explained.

Requested changes

See report above.

  • validity: -
  • significance: -
  • originality: -
  • clarity: -
  • formatting: -
  • grammar: -

Author:  Josephine Suh  on 2023-09-29  [id 4019]

(in reply to Report 1 on 2023-09-27)
Category:
remark
answer to question
reply to objection

We thank the referee for helpful questions, and hope our explanations below can be satisfactory.

Regarding the flow of logic in the paper and positioning of our work in relation to prior work, it may be helpful to clarify the following:

The goal in these two papers is to construct from scratch the two-dimensional JT theory involving the dilaton, purely within the confines of the quantum theory describing the boundary of the JT theory (particle moves in $AdS_2$, quantum observable is position of particle). In particular, we do not make use of the AdS/CFT duality and associated relations such as the Ryu-Takayanagi formula. Note these are best understood as UV/IR relations between quantities in a microscopic theory, and an effective gravity theory at low energies. In contrast, our starting point for constructing the gravity theory - that spacetime is the target space of a quantum observable, and the volume measure is a probability measure for it - are relations within the low-energy theory. As such, they are best understood not as "new entries in the holographic dictionary", but as newly identified relations (that we conjecture should hold generally) between spacetime and corresponding quantum degrees of freedom, in a quantized (in the bottom-up sense) theory of gravity.

To address the referee's individual points:

I also could not see what the conceptual leap in this paper is compared to the first one (except for Markovian to local Markovian) and hence cannot make up my mind about its suitability in SciPost. Currently, I feel it is better suited for SciPost Core.

The significance of a generator for a classical Markov process is that it exponentiates to the action of conditional probability $\mu_{T_2, T_1}(x_2, x_1)$ on measures, and in doing so, characterizes the entire process, as Markovianity implies joint probabilities of higher-order are obtained by repeated application of two-event conditional probabilities.

In the first paper, we worked with an asymptotic, flat limit of the stochastic process describing the boundary of JT gravity, and found the joint probability distributions it produces in the classical limit satisfy the precise Markov property above. This motivated us to examine (for the process at hand) the quantum analogue of the generator for a classical Markov process, and we found this leads to Einstein's equations in the flat limit.

Crucially, we relied on the Markov property, which in turn clearly depended (as seen from calculations) on going to the flat limit. Thus it was not at all clear that our method could be used to derive the full Einstein's equations in AdS including the cosmological constant.

In technical terms what we accomplish in this second paper, is i) to obtain an asymptotic limit of the exact stochastic process of the boundary of JT gravity distinct from the flat limit and corresponding to its dynamics at long times (see Figure 1), and ii) to show that the quantum "generator" of this process, using the same definition as if it were a Markov process, leads to Einstein's equations with the correct cosmological constant term.

These results motivated us to discuss a notion of "local Markovianity", roughly corresponding to the intuition that a process is "locally Markov" in time if it can be "stitched together" from patches of exact Markov processes. We believe such a notion of local Markovianity is novel and has not been discussed in the literature before. (Not surprising, as it is motivated by making a connection from stochastic processes to spacetime manifolds, which has not been done before.) We have not fully quantified this property in our current work, but believe doing so will be an important task in the future with regards to characterizing which stochastic processes give rise to gravity.

In short, this second paper contains crucial technical and conceptual leaps that allow us to put forward the conjecture that spacetime should be identified as probability from the quantum point of view, and that "general relativity arises from quantum stochastic processes".

1. In the introduction, second para, the author makes it seem that she is the first one to try and address this issue of the quantum information theoretic interpretation of Einstein's equation. This has already been tackled in numerous papers in the literature and goes by the name of 1st law of entanglement. The author should cite these original papers and attempt explain in what manner her proposal hopes to distinguish and lead to new insights compared to these older works.

First, we hope our first paragraph above makes it clear that our work is quite disparate from previous attempts to derive Einstein's equations in the bulk using AdS/CFT duality. To emphasize, we do not use the AdS/CFT duality or the Ryu-Takayanagi formula in any way; our goal having been to situate quantum gravity as a completely quantum phenomenon, without making use of any prior "dualities" between quantum theory and gravity.

Another way to view this is as follows: any attempts to derive Einstein's equations from the RT formula was bound to have limits due to the fact that the RT formula is a relation between codimension-two surfaces and an information-theoretic quantity in a quantum theory, whereas Einstein's equations are a local statement. (In the case of work using the first-law that the referee mentions, these limitations manifest in the fact that the derivation applies only in the CFT vacuum or certain conformal transformations of it; it is only then that one has sufficient symmetry.)

What we achieve in our work, is to attach an information-theoretic significance to the volume measure at each point of spacetime. It turns we are able to do this in a low-energy context that is subtly different from the UV-IR setup of AdS/CFT duality. (It is an interesting question to try to "lift" our results to the UV-IR setup of AdS/CFT duality, we are trying to address this in current work in progress.)

2. End of page 2, it is stated that mostly such kernels will be considered which depend only on time differences. Will this automatically restrict to theories with time translation symmetry? [and hence exclude cosmological scenarios?]

In our calculations in this paper we only considered thermal (and therefore time-translation invariant) states explicitly. However, even in cases without time-translation symmetry, a generator can be defined at each time.

3. The spectrum of the bulk theory should consider a spin-2 massless particle. How does one think of higher spin theories in the language of probability distributions?

This is an interesting question; we believe it will be best addressed with further work.

4. Section 2 is titled Generators of non-Markovian process. I don't see any generators being given in this section explicitly.

The definition of the generator is the same as if the process was Markovian, $G_{T_1}\nu=\lim_{T_2 \to T_1}\partial_{T_2}\int\mu_{T_2, T_1}\nu$. The point is that as long as one is considering the evolution of single-event distributions (instantaneous characterization of process), it is okay to pretend that the conditional probabilities of the non-Markovian process compose as if it was a Markovian process. See eq. (8). However, this approximation fails at the level of higher-order distributions.

5. While I see -16 in equation 38, I struggled to see what exactly the difference was compared to the first instalment that enabled one to get the cosmological constant.

As discussed above, we had to obtain an asymptotic limit of the stochastic process of the boundary of JT gravity distinct from the flat asymptotic limit used in our first paper, and which corresponds to dynamics at time scales long compared to the AdS radius. See illustration in Fig. 1. The results are detailed in Section 3.1.

6. Apart from the interesting first sentence in the abstract, I did not feel that the connection with Ryu-Takayanagi was clearly explained.

As discussed above, the RT formula for us primarily served as motivation. We did not make any use of it in our work. It is an interesting question for the future, for example, to use our work to derive the RT formula.

---

## Round 2 · Referee Report · Anonymous (Referee 2) · 2023-11-8

Strengths

  1. ambitious conjecture of quantum derivation of Einstein equations and the paper proves this conjecture for a certain regime in two dimensional dilaton gravity.

Weaknesses

  1. Hard to read, very long sentences making hard to follow the logic, specially in the intro

Report

This is the second paper of a series of two. The novel idea is to connect the volume measure of spacetime to be a probability measure constrained by quantum dynamics and Einstein's equations to evaluation of a probability under a quantum process. This second paper studies the anti de Sitter limit of the results which requires a generalization of Markovian property.

I'm not an expert on these questions but I found the idea original and well motivated. I would nonetheless ask the author to review the introduction to clarify the logic (I make some examples of suggestions below).

I would recommend the paper for publication (after taking the comments into account) as the first paper on the same series was published in SciPost and the results in this preprint require the development of new tools that go beyond the case studied in the first paper.

Requested changes

  1. In the introduction, please add the definitions of symbols,letter used in formulas. It makes it hard for a non familiar reader to understand the formulas. Example: what is T12 in (4), what is \Phi?
  2. In the introduction, page 2, the last paragraph starts by a question; what is the connection to gravity... ? I didn't understand where the author was answering this question.
  3. Pg 6, the author discusses the holographic limit. Just after eq. 10, it is associated with \gamma and L large. But around es. (14) the information on L has disappeared. What L becomes in this limit?
  4. Pg 7, what is T_b in figure 1?

---

## Round 3 · Referee Report · Anonymous (Referee 1) · 2023-11-24

Report

The paper has improved enough to be accepted for publication in SciPost.

---

## Round 3 · Referee Report · Anonymous (Referee 2) · 2023-12-11

Report

I thank the author for making the improvements and the corrections. I recommend the paper for publication.

---

## Round 3 · Author Response

Dear Editor,

We have revised the manuscript to take into account the referee's feedback and suggestions.

Best regards,
Josephine

---

## Round 3 · List of Changes

1. We have rewritten parts of the introduction, in particular the last three paragraphs on p. 2, to clarify the logic and explain the analogy to chaos better. We have also tried to shorten sentences when appropriate.

  2. On top of p. 17, we have added a paragraph about the relation of our work to known aspects of AdS/CFT including the Ryu-Takayanagi formula.

  3. We have made other small changes requested by referees, clarifying notation and adding definitions.

---

## Editorial Decision

published